# A fungal pathogen induces systemic susceptibility and systemic shifts in wheat metabolome and microbiome composition

Heike Seybold [1,2,6], Tobias J. Demetrowitsch [3,7], M. Amine Hassani [1,2,7], Silke Szymczak [4], Ekaterina Reim[1,3], Janine Haueisen [1,2], Luisa Lübbers[1], Malte Rühlemann [5], Andre Franke [5], Karin Schwarz[3] & Eva H. Stukenbrock [1,2✉]

Yield losses caused by fungal pathogens represent a major threat to global food production. One of the most devastating fungal wheat pathogens is *Zymoseptoria tritici*. Despite the importance of this fungus, the underlying mechanisms of plant–pathogen interactions are poorly understood. Here we present a conceptual framework based on coinfection assays, comparative metabolomics, and microbiome profiling to study the interaction of *Z. tritici* in susceptible and resistant wheat. We demonstrate that *Z. tritici* suppresses the production of immune-related metabolites in a susceptible cultivar. Remarkably, this fungus-induced immune suppression spreads within the leaf and even to other leaves, a phenomenon that we term "systemic induced susceptibility". Using a comparative metabolomics approach, we identify defense-related biosynthetic pathways that are suppressed and induced in susceptible and resistant cultivars, respectively. We show that these fungus-induced changes correlate with changes in the wheat leaf microbiome. Our findings suggest that immune suppression by this hemibiotrophic pathogen impacts specialized plant metabolism, alters its associated microbial communities, and renders wheat vulnerable to further infections.

[1] Botanical Institute, Kiel University, Am Botanischen Garten 1-9, 24118 Kiel, Germany. [2] Max Planck Institute for Evolutionary Biology, August-Thienemann-Str. 2, 24306 Plön, Germany. [3] Institute of Human Nutrition and Food Science, Kiel University, Heinrich-Hecht-Platz 10, 24118 Kiel, Germany. [4] Institute of Medical Informatics and Statistics, Kiel University, University Hospital Schleswig Holstein Campus Kiel, Arnold-Heller-Str. 3, 24105 Kiel, Germany. [5] Institute of Clinical Molecular Biology, Kiel University, Am Botanischen Garten 11, 24118 Kiel, Germany. [6]Present address: Alexander Silberman Institute of Life Sciences, The Hebrew University of Jerusalem, Givat Ram, Jerusalem 9190401, Israel. [7]These authors contributed equally: Tobias J. Demetrowitsch, M. Amine Hassani. ✉email: estukenbrock@bot.uni-kiel.de

Plant pathogens can be classified according to their lifestyle[1]. Biotrophic pathogens colonize and feed on living host tissue to complete their lifecycle, whereas necrotrophic pathogens can induce plant cell death and feed on nutrients released from dying host cells. Hemibiotrophic pathogens initially colonize hosts via biotrophic invasion and later switch to necrotrophic growth. Plant immune responses against biotrophic and necrotrophic pathogens differ considerably[2]. Immune defenses induced by biotrophic pathogens involve the accumulation of anti-microbial metabolites and local cell death conferred by a hypersensitive response[2,3]. Plant pathogens produce effector molecules to avoid or suppress immune responses[3]. While some effectors have evolved to avoid immune recognition by the plant, others protect the fungus from plant-derived apoplastic defense mechanisms or reprogram intracellular plant responses[4–7]. Plant defenses are often not specific to single pathogens, but generally target a broad range of microbes. As a consequence, different pathogens have evolved effectors targeting the same defense responses independently from one another[8]. Conversely, suppression of defense-related host responses by one pathogen may enable additional infections by other pathogens[9].

*Zymoseptoria tritici* is a global hemibiotrophic plant pathogen that infects wheat, causing up to 50% yield loss[10]. Resistance breeding in wheat can confer complete resistance to particular isolates of the fungus. For example, the resistance gene *Stb6* confers resistance against *Z. tritici* isolates expressing the fungal effector protein AvrStb6[11,12]. To date, 21 such resistance genes against *Z. tritici* have been described[13]. Still, farmers mainly rely on chemical control to prevent and manage the disease. In fact, 70% of fungicides in the European Union are used to control *Z. tritici*. However, fungicide resistance is an increasing problem[10,14]. One challenge in decreasing fungicide usage in the control of *Z. tritici* is a long biotrophic infection phase before the switch to visible necrotrophy. The infectious process is poorly understood, and few *Z. tritici* effector proteins have been identified and functionally characterized to date[5,15–18].

In this study, we looked at the interaction of *Z. tritici* with its host during the biotrophic stage of infection. We first addressed whether biotrophic fungal colonization of wheat involves active suppression of immune responses or if the pathogen only avoids host recognition. We also investigated the extent to which *Z. tritici* infection influences colonization of the plant by other microbes. If the fungus actively suppresses immune responses in susceptible wheat cultivars, this could influence the ability of other nonadapted microorganisms to colonize the plant. To test our hypotheses, we conducted coinfection experiments of *Z. tritici* with adapted and nonadapted *Pseudomonas syringae* bacteria. Here, we provide evidence for active immune suppression in plant tissues both local and distant to the infection, a new effect that we termed "systemic induced susceptibility" (SIS). We applied a comparative Fourier-transform ion cyclotron resonance mass spectrometry (FT–ICR–MS) metabolomics approach to describe the observed differences on the wheat metabolome. In addition, we analyzed the bacterial microbiome to generalize the observations made during the coinfections.

## Results

### Resistance to *Z. tritici* is cultivar-dependent.
In compatible infections, *Z. tritici* propagates in the leaf mesophyll and later shifts to necrotrophic growth and pycnidia production[19]. In contrast, the fungus is not able to propagate in the mesophyll of incompatible host plants. In order to understand the underlying traits that define compatible versus incompatible interactions between different cultivars of wheat and the fungal pathogen *Z. tritici*, we conducted infection experiments with the susceptible wheat cultivar Obelisk (without the resistance gene *Stb6*) and the resistant cultivar Chinese Spring (with the resistance gene *Stb6*)[20]. The *Z. tritici* isolate IPO323 producing the effector protein AvrStb6 was used in all infection experiments. In agreement with previously published studies, *Z. tritici* isolates producing AvrStb6 are able to infect and reproduce in leaves of Obelisk, but infection is aborted in leaves of the resistant cultivar Chinese Spring[19,20] (Fig. 1a, b, Supplementary Fig. 1).

There are no immune markers established for hexaploid wheat, and we lack good predictors of wheat response to fungal infection in different cultivars. We therefore developed a new assay based on bacterial infection to test our hypothesis that *Z. tritici* suppresses immune responses actively in compatible interactions. We predicted that bacterial growth would benefit from *Z. tritici*-mediated immune suppression and therefore could serve as a read-out for the extent of plant immune response following fungal infection. First, we tested the susceptibility of the cultivars Obelisk and Chinese Spring to the *P. syringae* pathovars *oryzae* (*Por*36), *tomato* (*Pst* DC3000 and its T3SS mutant hrcC-), and *maculicola* (*Psm* ES4326) (Supplementary Fig. 2). While Obelisk and Chinese Spring differ in their susceptibility to *Z. tritici*, the two cultivars show the same extent of susceptibility and resistance to the *P. syringae* pathovars (Fig. 1c).

### Fungal infection promotes bacterial coinfection.
To assess the spatial and temporal impact of *Z. tritici* on the wheat immune response during biotrophic growth, we then coinfected the *P. syringae* pathovars *oryzae* (*Por*) and *tomato* (*Pst*) on distinct leaf areas of Obelisk and Chinese Spring: (1) in the same area as the fungal spores (local), (2) adjacent to the fungal infection on the same leaf, and (3) on a leaf other than the one infected by *Z. tritici* (a systemic leaf) (Supplementary Fig. 3a). Bacteria were inoculated on leaves 4 days after inoculation with fungal spores (4 dpi-f).

Local coinfection of Obelisk with *Z. tritici* and *Por* increases bacterial growth compared to a mock-treated leaf (Fig. 2a). The increase in bacterial growth on *Z. tritici*-infected leaves ceased at later stages of coinfection (Supplementary Fig. 4). We predicted that *Z. tritici* efficiently suppresses the immune response in Obelisk, thereby promoting the growth of *P. syringae*. In support of this hypothesis, we observed reduced growth of the bacteria in leaves of Chinese Spring in which immune responses are induced by *Z. tritici* (Fig. 2a). Local coinfection cannot exclude a direct effect of fungal-bacterial interaction. Therefore, our experiment also included measures of bacterial growth in adjacent leaf areas. We observed that fungal infection in Obelisk facilitated bacterial growth in adjacent leaf areas, while coinfected Chinese Spring became more resistant to bacterial infection in adjacent leaf areas (Fig. 2b). This finding supports a direct effect of *Z. tritici* on the plant immune system. We further tested heat-killed fungal spores to confirm that only viable *Z. tritici* cells actively suppress plant immune responses. We confirmed the induction of susceptibility for the nonadapted *Pst* pathovar, which, in contrast to *Por*, hardly grows on Obelisk in the absence of *Z. tritici* but strongly benefits from fungal coinfection (Fig. 2c). Coinfection with *Z. tritici* also enabled wild-type-like growth of the T3SS-defective *Pst* hrcC- in the adjacent tissue (Supplementary Fig. 5). A beneficial effect on bacterial growth was detected both distal (toward the leaf tip) and proximal (towards the leaf base), but the effect was more distinct in proximal tissue (Supplementary Fig. 6). We then studied the reach of immune suppression by *Z. tritici* in infected plants. When the third leaf was inoculated with *Por*, bacterial growth was increased in the susceptible cultivar and reduced in the resistant cultivar, providing further support for the occurrence of SIS in Obelisk and systemic acquired resistance (SAR) in Chinese Spring (Fig. 2d).

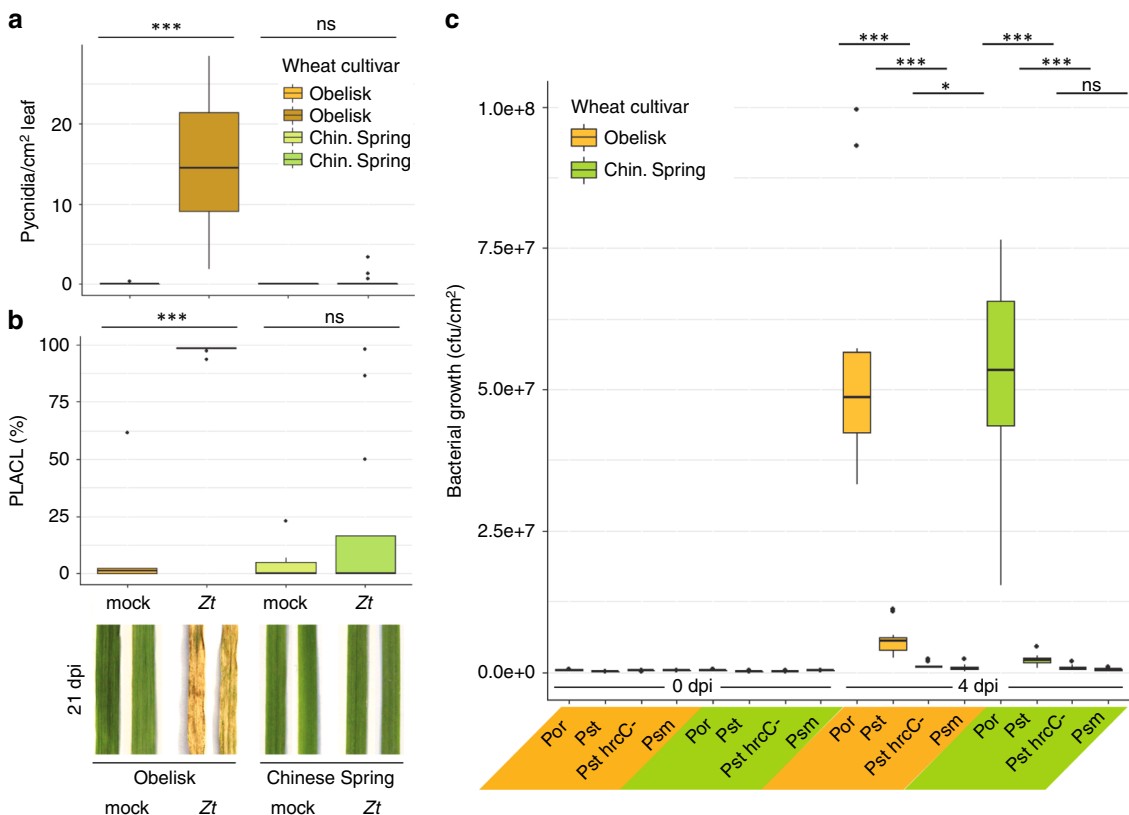

**Fig. 1 Contrasting resistance phenotype of wheat against *Z. tritici* but not *P. syringae*. a** Number of *Z. tritici* (*Zt*) IPO323 pycnidia per cm² leaf of wheat cultivar Obelisk (orange/brown) and cultivar Chinese Spring (green) at 21 dpi-f. **b** Percentage of leaf area covered by lesions (PLACL) as in **a** and representative leaf phenotypes at 21 dpi-f. **c** Bacterial growth of *P. syringae* pv. *oryzae* (*Por*)/pv. *tomato* (*Pst*)/pv. *maculicola* (*Psm*) at 0 days post bacterial infection (dpi-b) and 4 dpi-b in wheat cultivars as in (**a**). Statistical analysis was performed using a Shapiro–Wilk test of normality followed by a Wilcoxon rank-sum test of null hypothesis. *$P < 0.05$; ***$P < 0.001$. Number of biologically independent replicates: **a**, **b** Ob/CS mock ($n = 6$), Ob *Zt* ($n = 14$), CS *Zt* ($n = 16$); **c** 0 dpi-b ($n = 3$), 4 dpi-b ($n = 9$). Fungal and bacterial infections, respectively, were carried out at least twice with similar results.

***Z. tritici* induces systemic changes in the wheat metabolome**. We next asked which physiological responses of wheat confer *Z. tritici* susceptibility in Obelisk and resistance in Chinese Spring. In order to identify metabolites involved in the wheat immune response against *Z. tritici*, we conducted a suspected targeted metabolome analysis using FT–ICR–MS. We used infected and mock-infected leaves and analyzed leaf areas both local and adjacent to the infection (Supplementary Fig. 3b). Samples were taken on the same days post infection as in the measures of bacterial growth in the coinfection experiments (4 and 8 dpi-f). Overall, we measure 37,664 different peaks, of which 296 were annotated in a list of plant secondary metabolites (Supplementary Data 1). We focused on the annotated metabolites, especially those known to be immune-related metabolites and pathways.

First, we compared the complete metabolome dataset of Chinese Spring and Obelisk including unannotated metabolites and generated a principal component analysis plot to visualize the overall differences in the metabolomes of the two plant genotypes and in infected versus uninfected samples. The wheat cultivar and the leaf position with respect to the site of fungal infection explained the main differences within the dataset (Fig. 3a). By comparing mock- and fungal-infected leaf areas (local and adjacent to infection), we found that fungal infection did not cause a global shift in the metabolomes of Chinese Spring or Obelisk suggesting that only particular metabolites are targeted by the fungus (Supplementary Fig. 7).

We then set out to identify specific metabolites that accumulate differently in local and adjacent tissues to fungal infection of the two cultivars using the list of annotated metabolites

(Supplementary Data 1). We assessed the fold change of metabolite accumulation in three different comparisons. (1) In the cultivar comparison, an Obelisk sample was compared to a Chinese Spring sample (same treatment, same leaf position). (2) In the position comparison, we compared local samples to those from adjacent tissue (same treatment, same cultivar). (3) In the treatment comparison, samples infected with *Z. tritici* were compared to mock-treated samples (same cultivar, same leaf position) (Supplementary Fig. 8a, b). We identified 116 annotated metabolites with a significant difference in at least 1 of the comparisons (Supplementary Data 2). We refer to these metabolites as differentially accumulating metabolites (DAMs). We applied a Fisher's exact test to assess the significance of the observed differences in the three comparisons with respect to the complete dataset (Supplementary Table 1). Chinese Spring had significantly more DAMs than Obelisk (5191 with molecular formula annotated metabolites of 296,031 detected mass spectrometry peaks incidences in Chinese Spring compared to 3820 metabolites of 297,482 detected peaks incidences in Obelisk, $P < 0.0001$). We find a significant difference in the accumulation of DAMs in local and adjacent tissues of Chinese Spring compared to Obelisk (local: 4465 of 29,906 in Chinese Spring compared with 2466 of 27,914 in Obelisk, $P < 0.0001$; adjacent: 3390 metabolites of 27,374 peaks in Chinese Spring compared with 2474 metabolites of 30,945 peaks in Obelisk, $P < 0.0001$). In addition, we found significant differences between Chinese Spring and Obelisk for metabolites that were either upregulated (839 of 30,063 peaks in Chinese Spring compared with 397 metabolites of 28,248 detected peaks in Obelisk, $P < 0.0001$) or downregulated

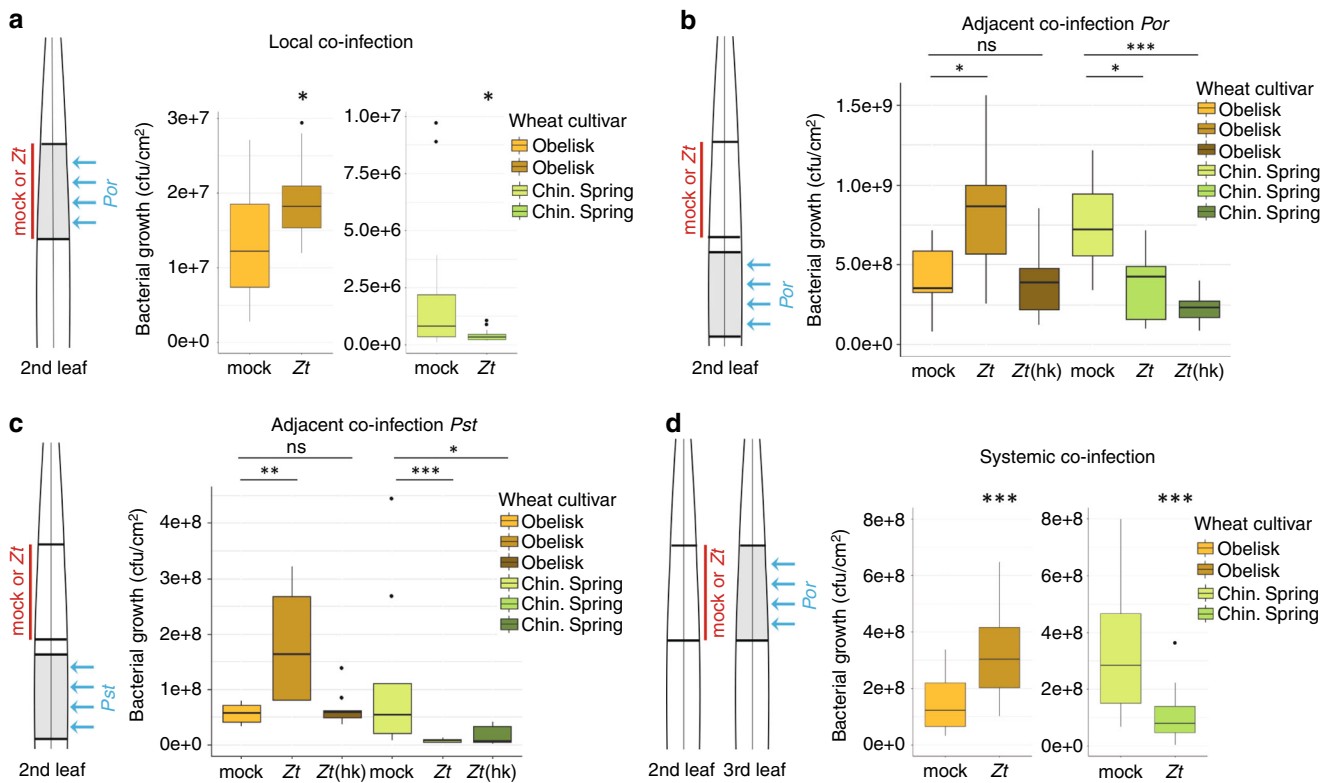

**Fig. 2 Infection of wheat with *Z. tritici* changes resistance to bacterial pathogens in local and systemic leaf tissues. a** Local coinfection (schematic) of wheat cultivar Obelisk (orange/brown) and cultivar Chinese Spring (green) with *Z. tritici* (*Zt*) IPO323 (red) on second leaf followed by *P. syringae* pv. *oryzae* (*Por*) (blue) on the same area of the leaf at 4 dpi-f and quantification of bacterial growth after an additional 4 days. **b, c** Adjacent coinfection (schematic) with *Zt* or heat-killed (hk) *Zt* locally (red) followed by *Por* (**b**) or *P. syringae* pv. *tomato* (Pst) (**c**) (blue) on an adjacent leaf area at 4 dpi-f and quantification of bacterial growth after an additional 4 days. **d** Systemic coinfection (schematic) of wheat cultivar Obelisk (orange/brown) and cultivar Chinese Spring (green) with *Zt* IPO323 (red) on the second leaf followed by *P. syringae* pv. *oryzae* (*Por*) (blue) at 4 dpi-f and quantification of bacterial growth after an additional 4 days. Bacterial coinfection took place on the third leaf. Statistical analysis was performed using the Shapiro–Wilk test of normality, which was then followed by the Wilcoxon rank-sum test for test of null hypothesis. *$P < .05$; **$P < .01$; ***$P < .001$; ns not significant. Number of biologically independent replicates: **a** Obelisk ($n = 12$), Chinese Spring ($n = 17$). **b, c** all treatments, $n = 9$. **d** Obelisk ($n = 24$), Chinese Spring mock ($n = 20$), Chinese Spring *Zt* ($n = 24$). Experiments were carried out independently at least three times (**a–c**)/twice (**d**) with similar results.

after fungal infection (584 of 24,672 in Chinese Spring compared with 584 of 29,060 in Obelisk, $P = 0.0058$).

***Z. tritici* targets immune-related biosynthetic pathways**. We studied *Z. tritici*-induced changes in entire biosynthetic pathways involving immune-related metabolites. We focused on two biosynthetic pathways that have been studied in other plant systems[21–29]: benzoxazinoids (BXs) and phenylpropanoids.

The BXs are a group of grass-specific antimicrobial phytoanticipines[30,31] (Fig. 3b). BXs are preformed and stored as inactive glycosides ready to be released as free BXs when needed[32]. Chinese Spring accumulates free BXs locally at 4 and 8 dpi-f (Fig. 3c, Supplementary Fig. 9), and inactive BX glycosides accumulate in adjacent tissue without direct contact with the fungal pathogen (Fig. 3d, Supplementary Fig. 9). In contrast, Obelisk does not accumulate free BXs or the storage forms upon infection with *Z. tritici*. We suspect that the presence of BX in Chinese Spring and the absence of BXs in Obelisk contribute to the resistance phenotypes of these wheat cultivars against the fungus.

Phenylpropanoids are a large class of phenolic secondary plant metabolites. Many stress-inducible phenylpropanoids and phenylpropanoid-derived compounds are categorized as phytoalexins against pathogens[29,33–35]. A comparative metabolome analysis revealed that *Z. tritici* has a strong effect on the biosynthesis of phenylpropanoids and compounds deriving from this pathway, such as hydroxycinnamic acid amides (HCAAs)

and flavonoids (Fig. 4). We found high levels of (hydroxyl) cinnamyl alcohols and glucosides of HCAs in Chinese Spring at 4 and 8 dpi-f in local and adjacent tissues of infected leaves. As (hydroxyl)cinnamyl alcohols are lignin precursors, we measured the lignin content in leaves of Obelisk and Chinese Spring prior to infection and three days after infection by *Z. tritici*. As the leaves grow and expand, lignin content in mock treated leaves increases in both cultivars (Supplementary Fig. 10). However, in accordance with our findings on lignin precursors (Fig. 4), we find a stronger increase in lignin content in Chinese Spring leaves infected with *Z. tritici* than in Obelisk. The reduced lignin content in fungal-infected leaves of Obelisk demonstrates how *Z. tritici* actively suppress an immune-related cell-wall strengthening in the susceptible host Obelisk (Supplementary Fig. 10).

In the comparative metabolome analysis, we also observe a delay in the accumulation of phenylpropanoid compounds and majority of flavonoids in Obelisk. Here, single phenylpropanoid and flavonoid compounds accumulated only at 8 dpi-f. Chinese Spring also accumulated HCAAs locally after fungal infection at 4 and 8 dpi-f. We found a significant increase in HCAA levels in Obelisk only in adjacent tissue at 4 dpi-f, and we detected no HCAAs at 8 dpi-f. These results indicate that fungal infection disturbs the biosynthesis of phenylpropanoids and related compounds in Obelisk.

Overall, we show that the regulation of immune-related pathways differs dramatically in response to *Z. tritici* infection in the Chinese Spring and Obelisk wheat cultivars. Immune-

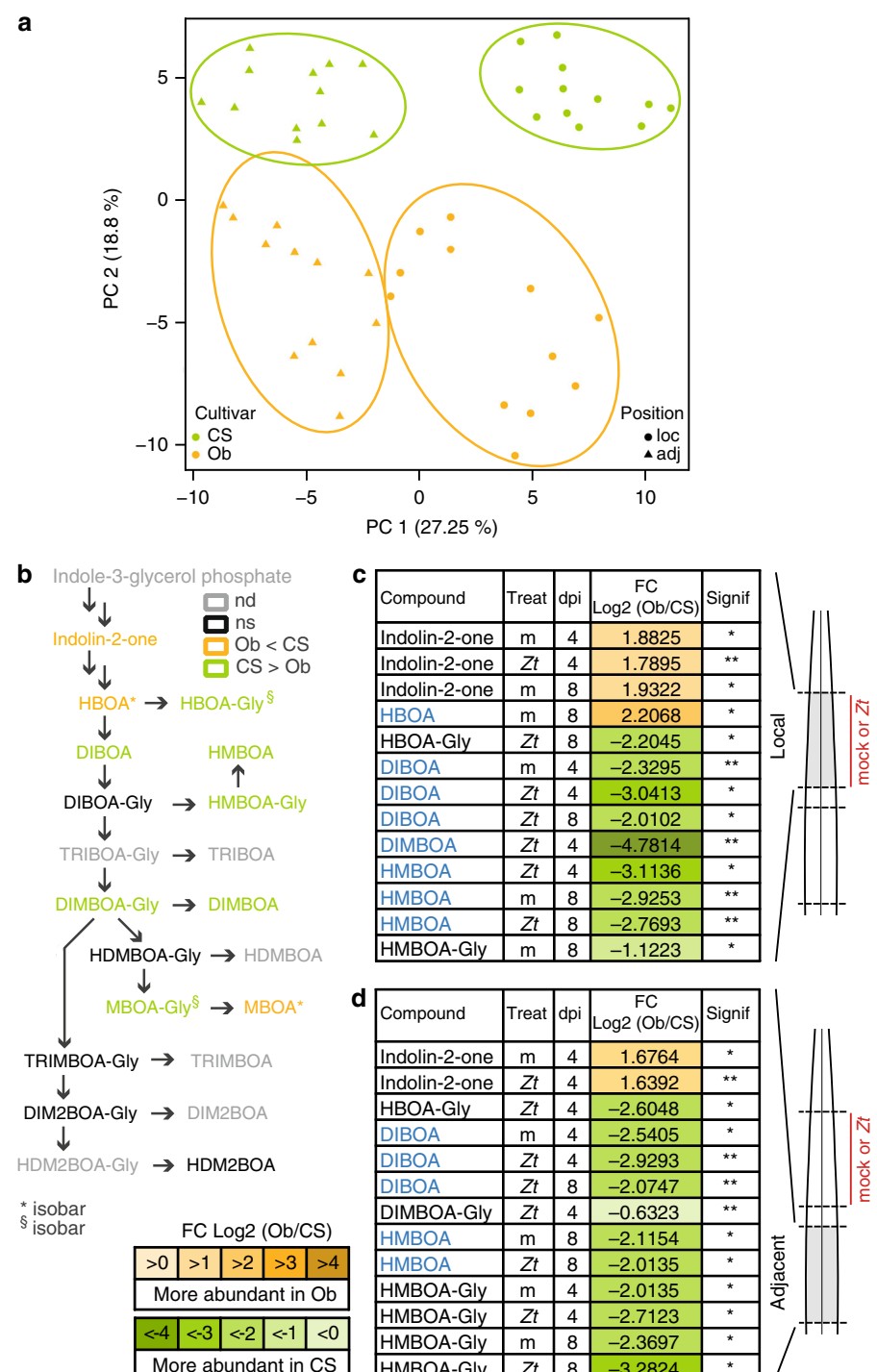

**Fig. 3 Biosynthesis of benzoxazinoids is altered locally and in adjacent tissues upon infection with *Z. tritici*. a** Principal component analysis (PCA) based on the complete metabolomics dataset. Samples are colored according to cultivar, and shapes refer to the position in the leaf. **b** Biosynthesis pathway of benzoxazinoids. Compounds with significant differences in the comparison between cultivars are highlighted in orange (Obelisk) or green (Chinese Spring). nd not detected, ns not significant. **c** Compounds of the biosynthesis pathway of benzoxazinoids with significant differences at local site of inoculation with *Z. tritici* (*Zt*) IPO323 or mock (m) at 4 and 8 dpi-f between wheat cultivars Obelisk and Chinese Spring. Active forms of benzoxazinoids are highlighted in blue. **d** Significantly different compounds in the benzoxazinoid biosynthesis pathway at an adjacent site to the inoculation in (**c**). Active forms of benzoxazinoids are highlighted in blue. Number of biologically independent replicates: *n* = 3. For details on the statistical analysis of the metabolomics dataset, please see the methods section. *$P < 0.05$; **$P < 0.01$; ***$P < 0.001$.

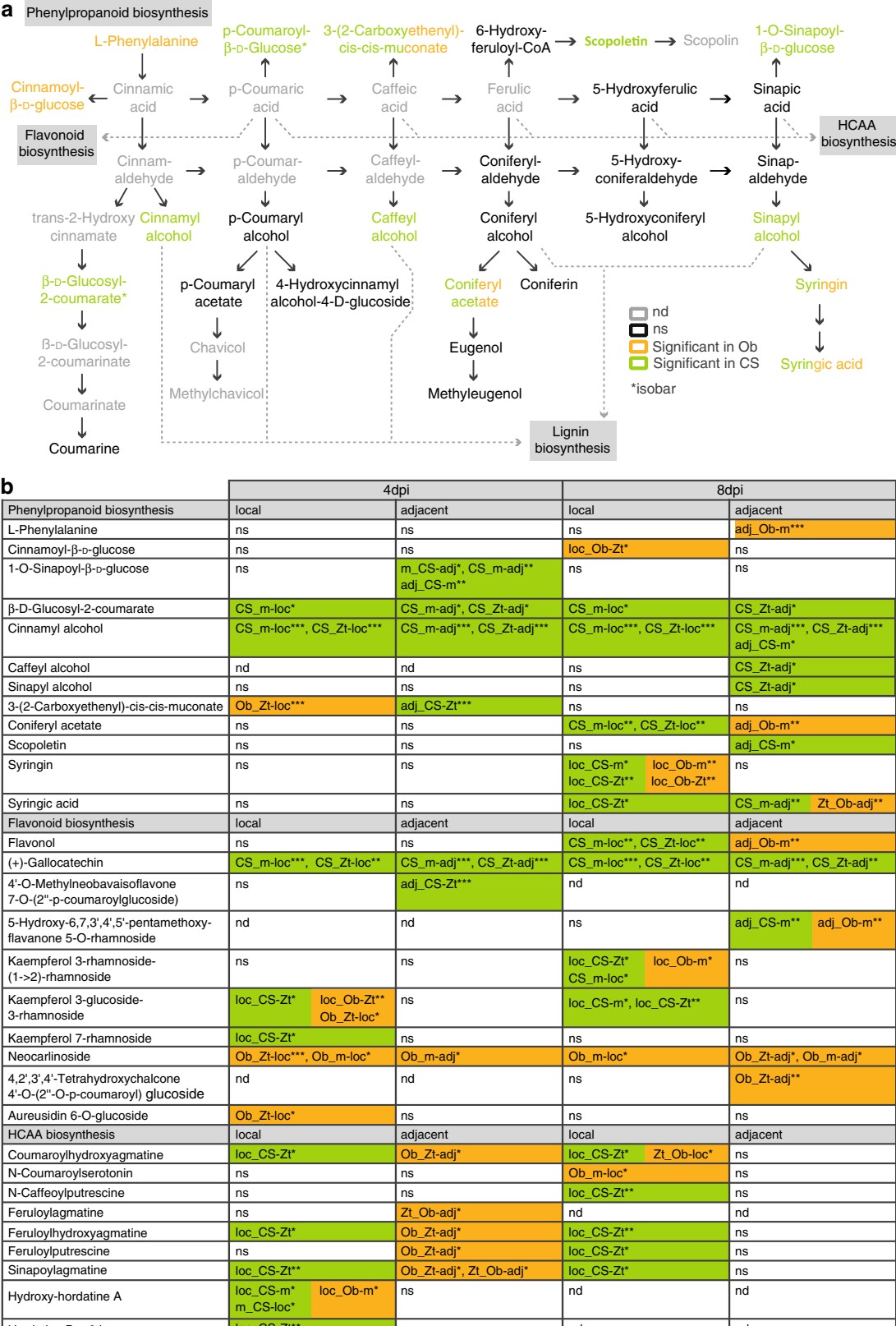

related DAMs are generally upregulated in the resistant cultivar Chinese Spring at the site of fungal infection and in adjacent tissues. We demonstrated that biosynthetic pathways and the accumulation of compounds involved in plant defense responses are manipulated by the infection of *Z. tritici* in the susceptible wheat cultivar Obelisk.

**Z. tritici infection correlates with systemic shifts in microbiome.** Above we show that *Z. tritici* suppresses immune-related biosynthetic pathways in susceptible wheat, ultimately enhancing fungal propagation. Moreover, the interaction of *Z. tritici* with the immune system of wheat in both the resistant and susceptible wheat has an effect on the growth of *P. syringae* locally as well as

**Fig. 4 Phenylpropanoids and branching pathways are differentially regulated in _Z. tritici_-resistant and _Z. tritici_-susceptible wheat cultivars.**
**a** Biosynthesis pathway of phenylpropanoids and branching biosynthesis pathways (modified from KEGG map00940). Compounds with significant differences in the comparison between treatments, cultivars, or leaf positions are highlighted in orange (Obelisk), green (Chinese Spring), or green/orange (Chinese Spring and Obelisk), respectively. nd not detected, ns not significant, Ob Obelisk, CS Chinese Spring. **b** Detailed information on metabolites with significant differences as highlighted in (**a**) and metabolites with significant differences from branching biosynthesis pathways. DAMs from the comparison "treatment" are named "m_" or "Zt_", DAMs from the comparison "cultivar" are named "Ob_" or "CS_", DAMs from the comparison "position" are named "loc_" or "adj_". Details on the labeling can be found in Supplementary Fig. 8c. Number of biologically independent replicates: $n = 3$. For details on the statistical analysis of the metabolomics dataset, please see the methods section. $*P < 0.05$; $**P < 0.01$; $***P < 0.001$.

systemically. Based on these findings we next hypothesized that infection by _Z. tritici_ and the corresponding changes in immune-related metabolites could have a more general effect to alter the composition and structure of plant-associated microbiota in local and adjacent leaf tissue. To address this question, we profiled the bacterial communities in Obelisk and Chinese Spring leaves after _Z. tritici_ infection at the same timepoints for which we monitored _P. syringae_ growth and for which we harvested metabolites for metabolome analyses. However, for the microbiome profiling we also included 0 dpi-f (Supplementary Fig. 3c).

We sequenced leaf-associated bacterial communities (V5–V7 regions of the 16S rDNA) and identified 3139 bacterial operational taxonomic units (100% OTUs). To assess the effect of _Z. tritici_ infection on both local and adjacent leaf tissues, we computed two alpha diversity measures (Shannon Index and observed OTUs, Fig. 5a) for both wheat cultivars (Chinese Spring and Obelisk). Remarkably, we found that fungal infection correlates with a significant reduction in community richness at 4 and 8 dpi-f in local leaf tissues and at 4 dpi-f in adjacent leaf tissues in Chinese Spring. In contrast, _Z. tritici_ infection had no significant effect on the community composition of either local or adjacent leaf areas in the susceptible cultivar Obelisk (Fig. 5a). To further analyze how infection by _Z. tritici_ impacted the leaf bacterial community structure, we computed Bray–Curtis distances between samples and applied principal coordinates analysis (Fig. 5b, Supplementary Fig. 11). As expected, _Z. tritici_ treatment induced a shift in the community structure of both leaf tissues in Chinese Spring (Fig. 5b, Supplementary Table 2) that is mainly explained by the depletion of several Actinobacteria and Proteobacteria OTUs at 4 dpi-f in local leaf tissue and the enrichment of diverse bacterial OTUs at 8 dpi-f (Fig. 5c, Supplementary Fig. 12). Although the infection had no significant effect on the bacterial community composition of Obelisk leaves (Fig. 5a), it induced a significant shift in the community structure of both local and adjacent leaf tissues of Obelisk at 4 dpi-f (Supplementary Fig. 11, Supplementary Table 2). These data indicate that leaf-associated bacteria are strongly altered by _Z. tritici_ infection in a resistant wheat cultivar at early time points. We speculate that this is an indirect effect of upregulated immune-related metabolites and reflect SAR in the plant. Contrarily, in Obelisk the microbial diversity is little affected by fungal infection reflecting the reduced change in metabolites and the occurrence of SIS.

**SIS may promote _Z. tritici_ dissemination**. The increased growth of different _P. syringae_ pathovars provides evidence for highly efficient immune suppression of _Z. tritici_ during biotrophic colonization, an effect that spreads into systemic plant tissues (Fig. 2). We considered the biological relevance of this phenomenon, because bacterial proliferation also may imply competitors in the wheat tissues. One possible scenario is that _Z. tritici_ induces SIS to facilitate systemic infection by new _Z. tritici_ spores. _Z. tritici_ is propagated both sexually and asexually during the growing season. The asexually produced pycnidiospores are splash-dispersed upwards from leaf to leaf[36]. We have previously

shown that _Z. tritici_ isolates can differ dramatically in the timing of disease development, not only between isolates but also between spores of the same cultivar[19]. This variability in spore germination and host penetration may be adaptive if early colonizers can facilitate the infection of late colonizers by suppressing immune responses in systemic leaf areas.

To test this hypothesis, we set up an experiment in which we coinfected the systemic third leaves of Obelisk seedlings with _Z. tritici_ spores 4 days after _Z. tritici_ infection on the second leaf (Supplementary Fig. 3d). In addition to increased bacterial growth, we expected to see a similar effect on _Z. tritici_ development on the third leaf, namely an increased pathogen proliferation. We quantified the fungal biomass in the systemic leaf over a time course of 16 days using a quantitative polymerase chain reaction (PCR) assay because fungal biomass correlates with fungal success. In addition, we quantified the development of necrosis and pycnidia production in both the second and third leaves. Mock-infected second leaves served as the control for SIS. Visible symptom development on the third leaf started from 10 dpi onwards, but varied strongly between replicates (Supplementary Figs. 13–15). The development of necrosis appeared to be independent of the treatment on the second leaf (Fig. 6a). Regarding pycnidia formation and fungal biomass on 16 dpi, fungal success on the systemic leaf was increased with infection on the second leaf at 16 dpi, although not significant at this stage of infection (Fig. 6b, c).

## Discussion

SAR is known to be a central component of the plant immune system[37–39]. The spread of immune signals from the site of infection acts to prime systemic plant parts against further pathogen infection. In this study, we demonstrated for the first time that systemic immune signaling can confer increased susceptibility in systemic tissues of plants infected by virulent pathogens. We termed this new phenomenon, SIS.

In our studies, we have shown increased growth of nonadapted _P. syringae_ bacteria in the wheat cultivar Obelisk during infection with the fungal pathogen _Z. tritici_ (Fig. 2a–d). Obelisk is susceptible to the _Z. tritici_ isolate IPO323, and biotrophic phase invasion is followed by necrosis and asexual sporulation[19]. We hypothesized that the increased bacterial growth in leaf tissues local and adjacent to fungal infection is conferred by the efficient suppression of immune responses by _Z. tritici_-produced effectors. Infection with heat-killed fungal spores provides further evidence for active immune suppression by fungal effectors in Obelisk, as the presence of denatured fungal elicitors from heat-killed spores was not sufficient to trigger SIS in this cultivar. _P. syringae_ proliferate to the same extent in Obelisk when treated with heat-killed fungal spores as when mock-treated (Fig. 2b, c). In contrast, we found evidence for _Z. tritici_-induced SAR in the resistant wheat cultivar Chinese Spring; _P. syringae_ bacteria grew less in leaf tissues local and adjacent to fungal infection when leaves were treated with _Z. tritici_ spores. (Fig. 2b, c).

Our hypothesis that fungal effector proteins confer SIS in Obelisk was further supported by the observation that the

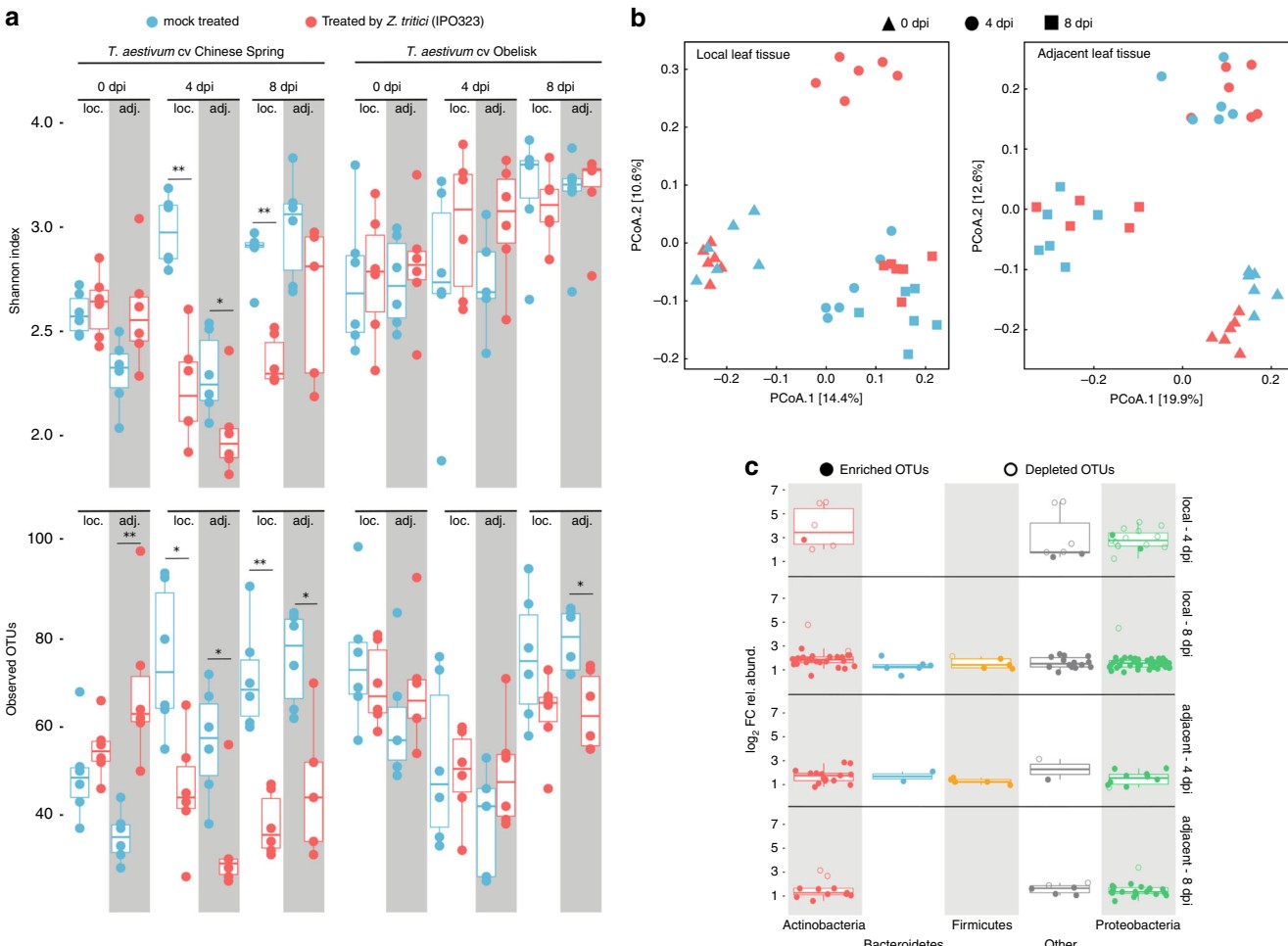

**Fig. 5 Infection by *Z. tritici* alters the wheat microbiota in local and adjacent leaf tissues. a** Two measures of the community composition, Shannon index and observed OTUs, are depicted in upper and lower panels, respectively. Reads were rarefied to an even sequencing depth corresponding to the smallest simple size of 601 reads. Community composition in leaf samples treated by *Z. tritici* (red boxplots) were compared to mock treatment (blue boxplots) for each corresponding time point (i.e., 0, 4, and 8 dpi-f) and leaf tissue types, local (loc.) and adjacent (adj.). Infection by *Z. tritici* leads to a significant drop in the computed Shannon indexes and observed OTUs in both types of leaf tissues (local and adjacent). Testing for significance was performed using a Kruskal–Wallis rank sum test. *P < 0.05; **P < 0.01; ***P < 0.001. **b** Principal coordinates analysis (PCoA) computed on Bray-Curtis distances of microbial communities associated with *T. aestivum* cultivar Chinese Spring. Each shape in the plot represents the community structure of one sample. Red- and blue-colored shapes designate mock- and *Z. tritici*-treated leaf samples as in (**a**). PCoA plots in the left and right panels correspond to local (left) and adjacent (right) leaf tissues, respectively. **c** Boxplot depicting log2 fold changes in the relative abundance of OTUs. Colors in the graph depict bacterial phyla. OTUs with P > 0.05 were filtered out. Filled and unfilled circles indicate OTUs that are significantly enriched and depleted, respectively, from *Z. tritici*-treated leaf samples compared to mock-treated samples. Number of biologically independent replicates: *n* = 6.

bacterial T3SS mutant strain *Pst* hrcC- grows like the wild-type *Pst* strain when coinfected with IPO323. The T3SS mutant is unable to secrete effectors and, in our experiment, relies on the immune suppression conferred by *Z. tritici* to grow (Supplementary Fig. 5). The wild-type-like growth of *Pst* hrcC in *Z. tritici*-infected Obelisk suggests that the fungal effectors partially replace the function of the absent *P. syringae* effectors and promote bacterial colonization. We also show that the effect of SIS is limited to the biotrophic infection phase of *Z. tritici*, as effects are reduced during later infection stages (Supplementary Fig. 4). Effector candidate genes of plant pathogens like *Z. tritici* show a dynamic expression pattern during the development of infection[3,19,40]. Therefore, the expression of effectors conferring SIS may be downregulated as the fungus develops into a necrotrophic lifestyle. In addition to fungal effectors, SIS could also be caused by yet to be identified fungal non-proteinogenic toxins with similar effects on plant immunity. Because we identified the induced susceptibility effect in adjacent and systemic tissues, we

excluded a nutritional effect of the fungus on bacterial performance, which would be limited to the infection site.

We investigated the metabolic components potentially underlying SIS. We compared local and systemic responses in wheat cultivars resistant and susceptible to fungal infection. We applied a suspected targeted metabolomics approach using FT–ICR–MS. Due to the ultrahigh resolution and the high sensitivity of this method, we identified a wide range of immune-related plant compounds. We have provided an overview of plant compounds and secondary metabolite pathways that are affected by *Z. tritici* infection in two wheat cultivars of differing susceptibility. We identified a variety of immune-related and antimicrobial plant compounds differentially produced in the resistant cultivar Chinese Spring, notably in local tissues after fungal infection. For example, we detected a salicylate conjugate at early stages after fungal infection specifically in the resistant cultivar (Supplementary Fig. 8d). Salicylic acid regulates local defense responses against biotrophic pathogens and is a crucial component of SAR

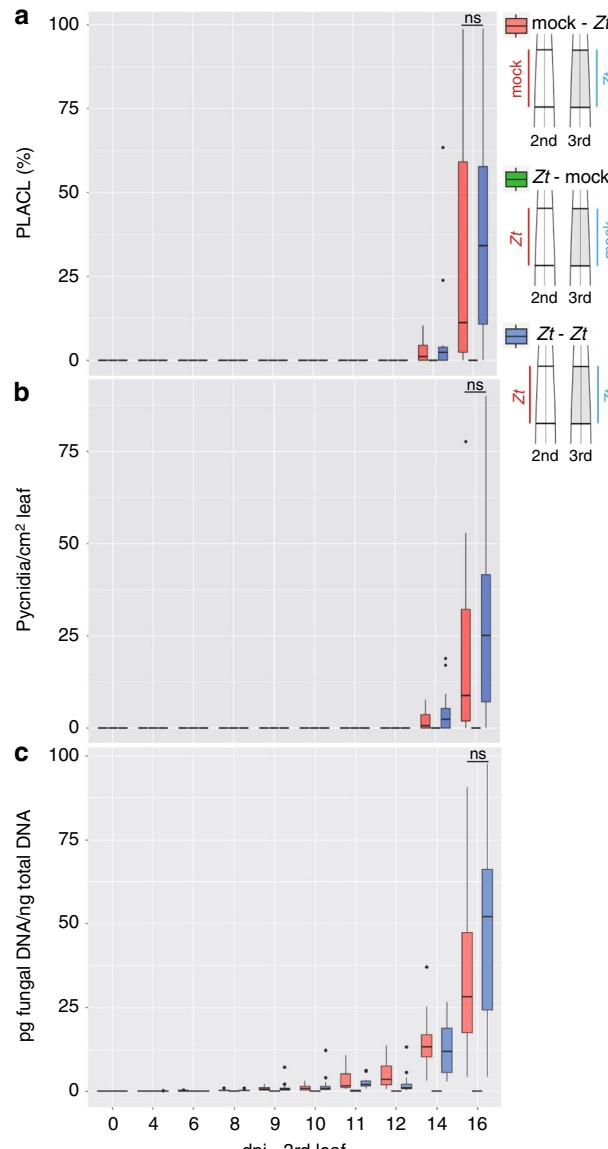

**Fig. 6 Local infection with _Z. tritici_ promotes systemic dissemination of the fungus. a** Percentage of leaf area covered by lesions (PLACL) throughout infection of cultivar Obelisk with _Z. tritici_ (_Zt_) IPO323. **b** Number of pycnidia of _Zt_ per cm² leaf of cultivar Obelisk as in (**a**). **c** The picogram (pg) fungal DNA/nanogram (ng) total DNA measured by quantitative real-time PCR. Statistical analysis was performed using the Shapiro–Wilk test of normality followed by a Wilcoxon rank-sum test for null hypothesis. ns not significant. Number of biologically independent replicates: _Zt_-mock (n = 3), mock-_Zt_ and _Zt_-_Zt_: 0–6 dpi (n = 12), 8–12 (n = 12), 14–16 (n = 12).

in both monocot and dicot plants[37–39,41]. The observed increase of salicylate conjugate levels confirms the local and systemic activation of defense responses against a biotrophic pathogen in Chinese Spring.

The phenylpropanoid pathway acts as a central hub for the biosynthesis of many immune-related compounds. In Chinese Spring, we found increased levels of several lignin precursors, such as alcohols of (hydroxy)cinnamic acids and diferulic acid (Fig. 4b, Supplementary Fig. 8d). These phenolic compounds are well known for their antimicrobial and antioxidant properties and support cell wall reinforcement after pathogen infection[42–45]. In accordance to this, total lignin content increased in Chinese Spring 3 days after infection with _Z. tritici_ (Supplementary

Fig. 10). Interestingly, we found a decrease in lignin content after fungal infection in Obelisk supporting the hypothesis that SIS negatively affects the biosynthesis of lignin precursors (Supplementary Fig. 10). Similarly to lignin precursors, flavonoid compounds as gallocatechin and kaempferol rhamnoside accumulate mainly in the resistant cultivar and are known to act as antimicrobials and antioxidants[25,35,46,47]. Accumulation of HCAAs is associated with resistance to various filamentous plant pathogens[23,24,27,48]. Interestingly, these compounds accumulated in leaf tissues adjacent to infection in the susceptible cultivar Obelisk at early stages of infection, whereas HCAAs were mainly increased locally in the Chinese Spring cultivar (Fig. 4b). Accumulation of HCAAs in adjacent tissues of Obelisk but not Chinese Spring might point to an overreaction of the HCAA pathway in the susceptible cultivar due to local _Z. tritici_-mediated inhibition of this pathway. Future studies should experimentally assess the functional role of this differential accumulation of HCAAs to validate the observed differences between the two wheat cultivars in response to _Z. tritici_ infection.

A second important pathway activated during fungal infection in many _Poaceae_ species is BX biosynthesis[32]. BXs are released from constitutively stored precursors following microbial invasion attempts. The antimicrobial effect of BXs, such as DIMBOA, has been described previously in maize, wheat, and other grasses[22,31,49]. In contrast to free BXs, BX glycosides represent inactive storage forms. Accumulation of BX glycosides in Chinese Spring tissues adjacent to infection (i.e., not directly challenged by _Z. tritici_) (Fig. 3d) may reflect priming of the immune system in the resistant cultivar for further fungal infection[50,51]. Interestingly, BX biosynthesis is blocked at an early stage in Obelisk leaf tissues both local and adjacent to infection (Fig. 3b–d), which might contribute to the virulence of _Z. tritici_ in this cultivar. We speculate that microbial manipulation of plant biosynthetic pathways and antimicrobial plant compounds contributes largely to fungal success in leaf tissues. In maize kernels, the virulence gene _FUG1_ of _Fusarium verticillioides_ has an impact on DIMBOA biosynthesis[52], and many effector candidate genes are upregulated during early stages of _Z. tritici_ infection[19,40]. Using our metabolomics approach, we may be able to predict steps in metabolic pathways that are potentially manipulated by such fungal effectors. This also suggests a large diversity in the effector repertoire and potential redundancy in immune manipulation.

Plants utilize an extensive spectrum of secondary metabolites to defend against attacking pathogens. Depending on the individual properties of such compounds, plants may be indirectly protected against other pathogens. However, these compounds may also influence the composition of other microbes systemically. For example, antimicrobial DIMBOA can impact the composition of the maize rhizosphere microbiome[53]. DIMBOA attracts _Pseudomonas putida_ (which is beneficial for the plant) to the maize rhizosphere, which shows increased tolerance towards the antimicrobial compound. In our study, we observed a strong shift in the microbial communities in the Chinese Spring cultivar (Fig. 5a, b). Because growth of _P. syringae_ bacteria was reduced in Chinese Spring during _Z. tritici_ infection (Fig. 2a–d), it is likely that other bacteria are impacted by SAR; most OTUs have reduced abundance in Chinese Spring during early _Z. tritici_ infection (Fig. 5c). At later times, this effect is reversed. We assume that the persistent microbial taxa can tolerate the accumulating immune-related compounds in Chinese Spring as shown for _P. putida_ in the maize rhizosphere[53].

Systemic changes in rhizosphere community structure due to activation of plant immune responses are not a new phenomenon[54,55]. Plants are thought to respond to an infection, such as through the recruitment of beneficial microbes from the soil to increase resistance[56,57]. Our unique approach of

combining coinfection studies with microbiome analysis, points to an active role of Z. tritici in shaping the microbiome structure in a susceptible host. Experimental validation of future studies (e.g., using fungal effector mutant strains) will confirm the direct or indirect role of Z. tritici on microbial community structure. As the effector set can differ between individual isolates of Z. tritici the effect of fungal infection on the microbiome composition may also depend on the individual isolates and their repertoire of effector proteins.

Finally, we considered the biological relevance of the observed SIS. Increased proliferation of other microbes could eventually be harmful to Z. tritici by increased competition in the phyllosphere environment. However, induced susceptibility in systemic leaf tissues could be a mechanism to promote infection by new Z. tritici spores. To test this hypothesis, we designed an experiment based on coinfection of asexual spores on different leaves of wheat seedlings (Fig. 6). While we did not observe a major effect, our results indicated that infection of the second leaf increases the efficiency of subsequent fungal infection of the third leaf. We selected timepoints for analysis that corresponded to those used for bacterial coinfection. However, it is possible that these time-points reflect minor effects of fungal coinfection. Z. tritici infection is a continuous rather than synchronous process. Z. tritici can persist as spore and hyphae on the leaf surface for up to 10 days before penetrating through stomata[58]. In addition, Z. tritici isolates display high variability in disease development[19]. Because infection in the field typically involves multiple Z. tritici strains[59,60], strains infecting later could mediate SIS in favor of already released pycnidiospores of fast colonizers. This would benefit the subsequent generation of fast-maturing pycnidios-pores during the development of individual plants[36]. In addition to the production of asexual pycnidiospores, SIS may also pro-mote sexual mating. Z. tritici has a heterothallic mating system, but both mating types (Mat1-1 and Mat1-2) can act as either male or female in sexual crosses. Previous studies demonstrated that avirulent Z. tritici isolates can mate with virulent strains but the avirulent strain will exclusively act as the male partner[12]. SIS may promote the coexistence of avirulent and virulent mating-compatible strains on the same leaf and thereby play a role in the outcrossing efficiency of Z. tritici.

Our findings emphasize the relevance of systemic dynamics that take place at various levels during plant infection. These dynamics include effects on the plant metabolome, but also organismic interactions at the microbial community level. Thus, further knowledge is needed to understand the interaction of the plant metabolome with the plant microbiome. Progress in this field is crucial for the development of future crop protection strategies based on plant probiotics and plant health-promoting microbes.

## Methods

**Wheat infection assays.** *Triticum aestivum* cultivars Obelisk (obtained from Wiersum Plant Breeding BV, the Netherlands) and Chinese Spring (kindly pro-vided by Bruce McDonald, ETH Zurich, Switzerland) were used for all infection experiments. Plants were grown in phytochambers on peat under constant con-ditions (16/8 h light (~200 $\mu$mol m$^{-2}$ s$^{-1}$)/dark cycle, 20 °C, 90% relative humidity).

The Z. tritici isolate IPO323 (kindly provided by Gert Kema, Wageningen University, the Netherlands) was used in all fungal infection experiments. Plants were infected with a spore concentration of $1 \times 10^7$ cells mL$^{-1}$ as previously described[19]. For the systemic fungal coinfection experiment, local infection with Z. tritici on the second leaf was followed by coinfection with the same strain on the third leaf 4 days after initial infection of the second leaf. The second and third leaves were harvested at the indicated timepoints. For further information on quantification of necrosis, pycnidia formation, and fungal biomass, detection of hydrogen peroxide and determination of lignin content, please see the Supplementary Methods.

The rifampicin-resistant *P. syringae* strains *Por36_1rif* (kindly provided by Henk-Jan Schoonbeek, John Innes Centre, UK), Pst DC3000, Pst DC3000 hrcC-,

and Pma ES4326 (kindly provided by Tiziana Guerra, Leibniz-Institut für Gemüse-und Zierpflanzenbau, Germany) were used for bacterial (co)infection experiments. Analysis of in planta growth of P. syringae in wheat was adapted from Schoonbeek et al.[61]. For wheat inoculation, bacterial suspension (OD$_{600}$ nm = 0.02) or control treatment was applied to the second leaf of 15-day-old wheat plants. The plants were sealed in a plastic bag and incubated for 4 days in phytochambers. Bacterial growth was assessed by counting bacteria after extraction and serial dilution. A detailed description of bacterial infection and quantification of bacterial growth can be found in the Supplementary Methods. For local bacterial coinfection experiments, the area for bacterial infection was the same as the area previously infected with IPO323. For adjacent bacterial coinfection experiments, the labeled area for bacterial infection was separated from the fungal infection area by a 1-cm buffer zone (Supplementary Fig. 16). Systemic bacterial coinfection took place on the third leaf while fungal spores were applied to the second leaf. If not stated otherwise, fungal infection was done 4 days prior to the bacterial infection.

**Metabolomics profiling.** Leaf material (6-cm leaf area) was harvested at 4 and 8 dpi-f, weighed, deep-frozen, and then extracted with methanol/water (80/20, v/v). For cell distribution, a Precellys Tissue Homogenizer was used. For extraction and suspensions, ultrapure LC–MS solvents and water were used. After extraction, all samples were stored at −80 °C until measurement.

An FT–ICR–MS (7 Tesla, SolariXR, Bruker, Bremen, Germany) was used in the flow-injection mode (a HPLC 1260 Infinity from Agilent (Waldbronn, Germany) was used). Water/methanol (50/50, v/v) with 0.1% acetic acid was used as transport eluent. The samples were ionized with an electrospray ionization source (both modes) and with 2 methods, so the detection range was from 65 to 950 Da. The average resolution at 400 m/z was 600,000. The main instrument parameters were dry gas temperature (nitrogen) of 200 °C at 4 L min$^{-1}$; nebulizer 1 bar; time-of-flight time section 0.35 ms and quadrupole mass 150 m/z with an RF frequency 2 MHz; and detector sweep excitation power of 18%.

Data evaluation was conducted with DataAnalysis 5.0 and MetaboScape 4.0.1, both from Bruker (Bremen, Germany). Sum formulas were calculated based on the mass error and isotopic fine structure. To reduce false-positive results, the seven golden rules of Kind and Fiehn were used[62]. In addition, annotation was conducted with customized databases by suspected targeted methods. The database was created based on 15 plant-related pathways from KEGG[63–65], such as general secondary plant metabolism or phenylpropanoid biosynthesis (Supplementary Table 3).

All mz values with assigned sum formulas were exported and further processed with the statistical software R (version 3.4.2). For each mode (SM/UM, positive/negative), measurements of mz values assigned to the same sum formula were added, and metabolites with measurements in fewer than three samples were removed. To create a single metabolomics dataset, metabolite data of the four modes were merged. For metabolites that were available in several modes, the one with the smallest number of missing values, or if ambiguous the largest median, was kept. Metabolite measurements were log$_2$ transformed, and missing values were replaced by the limit of detection ($= \log_2(10^6)$). Metabolite measurements were compared separately between breeds, positions, and treatments, and each comparison was stratified by the three remaining factors (including day). A linear regression model was fit for each metabolite separately, and P values were corrected for multiple testing using the Benjamini–Hochberg procedure[66].

**Microbiota profiling.** Seeds of Chinese Spring and Obelisk were washed three times and germinated on filter paper prior to sowing in pots containing peat inoculated with soil slurry (for details, see Supplementary Methods). Plant growth and fungal infection were carried as described above. Leaf material (6-cm leaf area for both local and adjacent tissue) was harvested at 0, 4, and 8 dpi-f. Each leaf sample was washed three times and stored at −80 °C for downstream processing. For DNA extractions, samples were homogenized and pretreated with lysozyme and proteinase K for 5 min at room temperature. DNA was extracted according to the manufacturer's protocol and stored at −20 °C for MiSeq library preparation. Detailed information on sample preparation and DNA extraction can be found in the Supplementary Methods.

The 16S rRNA gene DNA library for Illumina sequencing was prepared through a two-step PCR-amplification protocol. The 16S rDNA regions were PCR-amplified in triplicate using the primer set forward 799F (AACMGGATTA GATACCCKG) and reverse 1193R (ACGTCATCCCCACCTTCC)[67]. To block plant mitochondrial DNA, 10 times the volume 799F or 1193R of the blocking primer (GGCAAGTGTTCTTCGGA/3SpC3/) was added to each reaction. Technical replicates were pooled, and leftover primers were enzymatically digested. Amplicons from the first reaction were PCR-barcoded over ten cycles using reverse Illumina compatible primers (B1–B120). The PCR replicates from the second reaction were pooled, cleaned, and extracted from agarose gel plugs. PCR products were quantified and pooled. The library was cleaned twice with the AgencourtAMPure XP Kit (Beckman Coulter, Germany) and submitted for DNA sequencing using the MiSeq Reagent kit v3 with the 2× 300 bp paired-end sequencing protocol (Illumina Inc., USA). A detailed description of the library preparation procedure is included in the Supplementary Methods.

The forward and reverse sequencing reads were joined and demultiplexed using Qiime2 pipeline (2018.2.0)[68]. PhiX and chimeric sequences were filtered using Qiime2-DADA2[69]. All scripts used for read preprocessing are available at https://github.com/hmamine/ZIHJE/tree/master/raw_data_preprocessing. For alpha diversity analyses, count reads were rarefed to an even sequencing depth based on the smallest sample size of 601 reads using the R package "phyloseq"[70]. For community structure analyses, count reads were normalized by cumulative sum scaling normalization factors[71] prior to computing Bray–Curtis distances between all samples. To test for significantly enriched OTUs, the data were fitted to a zero-inflated Gaussian mixture model[71], and OTUs with Benjamini–Hochberg[66] adjusted $P < 0.05$ were displayed.

**Reporting summary**. Further information on research design is available in the Nature Research Reporting Summary linked to this article.

## Data availability

A reporting summary for this article is available as a Supplementary Information file. The source data underlying Figs. 1, 2, and 6 as well as Supplementary Figs. 4, 5, 6, and 10 are provided as a Source Data file. Raw data from microbiota profiling is available in the NCBI BioProject database under https://www.ncbi.nlm.nih.gov/bioproject/PRJNA549447. Data generated and analyzed during the metabolome study are available from the corresponding author upon request.

## Code availability

The scripts used in the microbiome analysis are available in GitHub under https://github.com/hmamine/ZIHJE as described in detail in the Supplementary Methods section.

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

## Acknowledgements

We thank Suayib Üstün for critical reading of the paper, and Carla Krone, Anna Krützfeldt, Maja Schmidt, and Alexander Maliskat for technical assistance. The study was funded by CIFAR, and a personal grant to EHS from the State of Schleswig Holstein and the Max Planck Society. Microbiome research in the group of EHS is also supported by the DFG Collaborative Research Center "Function and Origin of Metaorganisms" (SFB1182).

## Author contributions

H.S. and E.H.S. designed the experiments. H.S., E.R., J.H., and L.L. conducted and analyzed the plant infection experiments. T.D. performed the metabolome measurements. S.S. statistically analyzed the metabolomics dataset. H.S., T.D., and E.R. analyzed the metabolomics dataset. M.A.H. isolated the microbiota and extracted DNA and analyzed the microbiome dataset. M.A.H., M.R., and A.F. conducted the microbiome sequencing. E.H.S. and K.S. supervised the projects. H.S., T.D., M.A.H., and E.H.S. wrote the paper with input from all authors.

## Competing interests

The authors declare no competing interests.
