## [Peer Review File · Nature Communications]

Reviewers' comments:

Reviewer #1 (Remarks to the Author):

The manuscript by Seybold et al. carried out comparative analysis of the interaction of *Z. tritici* in susceptible and resistant wheat using coinfection assays, comparative metabolomics, and microbiome profiling. *Z. tritici* was found to suppress the production of immune-related metabolites in a susceptible cultivar. The fungus-induced immune suppression appears to spread within the leaf and to other leaves. Defense-related biosynthetic pathways are suppressed and induced in susceptible and resistant cultivars, respectively. In addition, infection by the fungus also affects the wheat leaf microbiome.

Although the data presented are potentially interesting, they are almost exclusively descriptive and failed to provide mechanistic insights on how immunity to *Z. tritici* is regulated/achieved in wheat. The differences in *P. syringae* pv. *oryzae* growth between mock and *Z. tritici* treatment were marginal and the differences in systemic fungal growth between mock and *Z. tritici* treatment were even smaller, which make the biological meaning of these differences questionable.

Reviewer #2 (Remarks to the Author):

The field of plant pathology is changing, climate change, toxic pesticides and resistance challenge efforts. The authors studied the mode of action of the fungal wheat pathogens *Zymoseptoria tritici*. They found a new mechanism, which they term "systemic induced susceptibility". This was supported by a defense-related biosynthetic pathway that are suppressed and induced in susceptible and resistant cultivars, and the finding that that *Z. tritici* suppresses the production of immune-related metabolites in a susceptible cultivar. This is an interesting finding, logic in my eyes and well known from other microbe-host interactions.

The strength of the study is the comprehensive experimental design including coinfection assays, comparative metabolomics, and microbiome profiling to study the interaction of *Z. tritici* in susceptible and resistant wheat as well as the logic, hypothesis-driven experimental design.

The weakness of the study is that this new plant-pathogen interaction was shown only in this artificial system for one pathogen, even for one strain of the pathogen. Here, it needs more evidence that it occurs in the field or works for other pathogens too.

The pathogen induced strong shifts in wheat leaf microbiota. It is of value that these studies are integrated, and the extent of the shift shows how important this is. However, as presented these results are isolated, and should be more linked with the other results. Moreover, I would suggest, to look at the whole pathosystem from the microbiome perspective. The pathogen is a member of the microbiome as well. Qualitative data of the microbiota are missing!

The definition of plant pathogens in these categories biotrophic, hemibiotrophic etc. is a really anthropocentric view, which needs revision. In this manuscript I would not place it so central that *Zymoseptoria* is hemibiotrophic. It's the function of all fungi to degrade organic matter.

The title has to be changed – it is too broad, only one pathosystem was studied.

Discussion section would profit from shortening.

Reviewer #3 (Remarks to the Author):

The manuscript reports on the correlation between SAR and pathogenic fungi. The study shows very interesting results, a possibility for fungi to suppress plant immune system based on metabolomics data. I would like to recommend the publication of the manuscript.

One comment:

Line 123, 37,664 seems the number of MS peaks. Then, to say the number is for metabolites is rather doubtful. So, better to say it is the number of peaks or signals in the analytical system.

Reviewer #4 (Remarks to the Author):

The manuscript aims to investigate the interaction between the fungal pathogen *Z. tritici* and wheat. The work presents three related stories. In the first story, the authors showed that upon infection with the fungal pathogen *Zymoseptoria tritici* an induced susceptibility in adjacent regions of the leaf is triggered. Second, they showed that there are differences in the accumulation of metabolomes in two wheat cultivars, one susceptible and one resistant. The third story is based on a microbiome analysis of wheat leaves and demonstrates that in the resistant cultivar there is a reduction in the richness and changes in the structure of the bacterial community upon infection with the pathogen. The work presents a very original work and provides good quality data which is of significance to the field. However, the work lacks a general focus and sometimes it is difficult to understand the logic of the text. Although the three stories are novel and related, the connection between them is not clear in the manuscript.

The authors have a very novel and interesting finding. They demonstrate that *Z. tritici* triggers induced susceptibility. They call this phenomenon "systemic induced susceptibility" (SIS). The main goal of the work is to characterize SIS, but the analysis performed does not provide evidences for the consequences of SIS in the microbiome or the metabolites involved in this phenomenon. To achieve this, first of all, the comparisons in the metabolome should be performed between mock and infected plants. In the results and figures, it seems (although it is not very clear) that this is not the main comparison performed. Comparisons between the two cultivars is valid, but would not provide information on what are the compounds involved in SIS (and SAR). This is mainly due to the complex genetic background of both wheat cultivars and that SIS (and SAR) are induced upon infection, as shown in figure 2. In addition, changes in microbiome were only observed in the resistant cultivar, which points more towards the direction that SIS does not lead to changes in the microbiome. If there are evidences for SIS changing the microbiome, it should be explained in the text and figures. I suggest that either the focus of the manuscript switches towards the characterization of induced resistance by *Z. tritici* in wheat or that the data providing mechanistic understanding of SIS is highlighted.

More detailed comments can be found below:

1- The representation of the metabolome is not very clear. In figure 4b, it is very difficult to understand which comparison was performed based on the provided nomenclature. Based on the text, it seems that the main comparison was performed between both cultivars. If the hypothesis is that changes in the content of some metabolites lead to the observed phenotype upon infection by *Z. tritici*, then the authors should compare the metabolomic profile of infected and uninfected leaves. Most frequently the comparisons are between cultivars, which is also relevant, but not conclusive for SIS and SAR.

The authors had interesting results related to the content of BXs in resistant and susceptible cultivars. In the resistant cultivar they observed increased levels of the inactive forms in adjacent tissues of the infection by the fungus. However, this was not observed in the susceptible cultivar. The authors conclude from this data that the fungus might manipulate the host in order to prevent the induction of this compound. However, it could very well be that only in Chinese Spring these inactive forms are accumulated and not in Obelisk. Actually, this seems to be the case since in Fig. S9 there is no induction of the BXs, but CS has higher constitutive levels of them. Thus, their

function in SIS remains unclear and *Z. tritici* does not seem to have capacity to suppress their accumulation.

In addition, it is known that Chinese Spring basal resistance is not higher than in other cultivars, but this cultivar is specifically able to recognize an effector of some strains of *Z. tritici* and subsequently trigger an immune response that hinders the progression of the pathogen (by the way this is key information that should be described in the introduction). The accumulation of the metabolites investigated in this work could contribute to the resistance response, but if this is the case, there should be differences in CS between the Mock and Zt treatments in figure 3c,d and S9. Based on the provided data this does not seem the case.

In summary, I think that the authors should focus on differentially accumulated compounds upon infection in order to have more conclusive data on SIS (and also on SAR).

2- In the discussion, the authors indicate that the data showed that pathogens have an active role in manipulating the microbiome (lines 316-322) in a susceptible host. Although this is true for the co-infections performed with *Pseudomonas*, it is not the case for the global microbiome analysis performed, since there are no differences between mock and Zt treatments in cultivar Obelisk in the figure 5.

The main goal is to characterize the effect of SIS in the microbiome and based on the data presented, there is an effect on the bacterial community upon a resistance response against *Z. tritici*, but not in the compatible interaction. If there were observed changes in the susceptible cultivar, they should be shown in the results.

3- Finally, although in the results it is said that there are statistical differences in figure 6b and 6d at 16 dpi, it is not clear in the figure. Either the asterisks are missing or they are difficult to see. In case that there are no statistical differences, it cannot be concluded that induced susceptibility in systemic leaf tissues is a mechanism to promote further infections by *Z. tritici* and also the results section should be modified.

4- I think that there are some sentences that are not accurate in the discussion/results. For instance, the authors mentioned that the role of fungal effector proteins in SIS is demonstrated in the experiment of T3SS. For making this statement, first the authors need to clarify if the *hrcC* mutant grows more in the presence of the fungus. Based on the figure S5, there are no statistical differences between mock "Pst *hrcC*" and "Zt Pst *hrcC*". Therefore, the sentences "the increased growth of Pst *hrcC* in *Z. tritici*-infected Obelisks..." (line 254) and "Coinfection with *Z. tritici* also enabled growth of the T3SS-defective..." (line 110) are not correct. Even if this is the case, this experiment only highlights that effectors of Pst are not required, most probably because of a suppression of the immune response by *Z. tritici*, but it is not suggesting anything about the role of *Z. tritici* effectors in suppressing the immune response of wheat.

Minor comments:

- The results shown in figure 1a and 1b are already known in the literature, but it was not cited properly. The authors should highlight that this was already known and cite Brown et al. 2001.
- The title of figure S5 is very difficult to understand.
- In figure S7 there is a typo. It should say Obelisk.
- In the metabolome section, it is not clear if the analysis was performed for the annotated or for the global metabolome. In line 125 it is mentioned that only annotated metabolites will be analyzed, but in line 126, it is mentioned that the complete metabolome dataset will be analysed.
- In this analysis, it is striking to observe that there are differences in the mock between local and adjacent. Do the authors have any explanation for that?
- Figure 3c,d would be clearer if it will also indicate which are the active and inactive forms.
- In line 162, it is mentioned "We suspect that the absence of BXs in Obelisk is partly responsible for the increased susceptibility of this cultivar". However, since we know that the resistance of CS is due to the immune response triggered upon recognition of an effector protein, it is most probably the other way round that the presence of BXs is responsible for the resistance of Chinese Spring.

Related to this, it is mentioned that BXs are not accumulated in cv. Obelisk (line 161), but this is not what is shown in figure S9.

In any case, in order to conclude about SIS and SAR, the authors should compare infected and non-infected tissues. Only upon infection, they observed differential colonization by the bacteria. Thus, if these metabolites are having a role, they should be differentially accumulated upon infection and not between cultivars.

- The sentence written in lines 183 and 184 is not very accurate, since there are not enough evidences to highlight that *Z. tritici* manipulates immune-related biosynthetic pathways in susceptible wheat cultivars.
- The statistical analysis was performed correctly, but I miss information on how many times were the experiments performed independently. Specially for the infection assays.
- Line 409 in materials and methods. How were the seeds washed?
- The results section has frequently information that should have been provided in the introduction section.

Reviewer #1 (Remarks to the Author)

The manuscript by Seybold *et al.* carried out comparative analysis of the interaction of *Z. tritici* in susceptible and resistant wheat using coinfection assays, comparative metabolomics, and microbiome profiling. *Z. tritici* was found to suppress the production of immune-related metabolites in a susceptible cultivar. The fungus-induced immune suppression appears to spread within the leaf and to other leaves. Defense-related biosynthetic pathways are suppressed and induced in susceptible and resistant cultivars, respectively. In addition, infection by the fungus also affects the wheat leaf microbiome.

Although the data presented are potentially interesting, they are almost exclusively descriptive and failed to provide mechanistic insights on how immunity to *Z. tritici* is regulated/achieved in wheat.

Reply: We note that analyses of immune responses in wheat are challenging given the lack of proper immune markers and the complex genetics of wheat with a hexaploid genome. Our attempts to use the reference genome of Chinese Spring (a landrace) for mapping of sequence reads from Obelisk (an inbred European variety) failed. Therefore, we focused on immune responses that do not rely on genomic sequences: Metabolic pathways involved in immune response are highly conserved among plants and metabolites – in contrast to genes – are very unlikely to differ between two wheat cultivars.

We are aware that this model system limits the possibility to mechanistically explain the observed differences. On the other hand, we are describing immune responses with immediate effects, as metabolites, in contrast to transcribed genes, directly can reveal antifungal/antimicrobial properties. We believe our findings at the metabolome level are not only novel, but also spectacular by providing a hitherto unprecedented overview of plant secondary metabolism as it is manipulated by a fungal plant pathogen.

The differences in *P. syringae* pv. *oryzae* growth between mock and *Z. tritici* treatment were marginal and the differences in systemic fungal growth between mock and *Z. tritici* treatment were even smaller, which make the biological meaning of these differences questionable.

Reply: We agree that compared to experiments done in the classical *Arabidopsis* – *P. syringae* pv. *tomato* system, the bacterial growth that we report in the wheat – *P. syringae* pv. *oryzae* system is lower. However, we would like to underline that we are using a highly different plant system that cannot be compared in this way to the well-established *Arabidopsis* system. The few published experiments that have used wheat and *P. syringae* pv. *oryzae* show bacterial growth which are comparable, in every way, to our results (see Schoonbeek *et al.*, *New Phytologist*, 2015). Moreover, if the bacterial growth is lower in wheat compared to *Arabidopsis*, we still expect differences between treatments to reflect actual differences between wheat cultivars, time points and leaf tissues. We note again, that the wheat system is less potent for this type of assay compared to *Arabidopsis*. Yet, since we are studying a fungal pathogen highly specialized on wheat, research of this pathogen relies on experiments in wheat and not *Arabidopsis*.

Since we cannot directly compare the results from our plant – pathogen system to the *Arabidopsis* system and results reported from this model system, we rely on careful statistical tests. Statistically, we can thus validate the relevance of the observed differences between treatments and wheat cultivars.

The biological relevance of a result should not be based on the difference observed in one type of experiment. To determine the biological relevance of our observation (that the fungal pathogen suppresses the wheat immune responses and therefore indirectly helps the bacteria grow) we assessed the fungal impact on the wheat immune responses at different levels. We thereby report the effect of fungal infection also on the plant metabolome and on the microbiome of wheat. The combination of these experiments is able to place the results from the bacterial growth assay into a larger biological context.

Reviewer #2 (Remarks to the Author)

The field of plant pathology is changing, climate change, toxic pesticides and resistance challenge efforts. The authors studied the mode of action of the fungal wheat pathogens *Zymoseptoria tritici*. They found a new mechanism, which they term "systemic induced susceptibility". This was supported by a defense-related biosynthetic pathway that are suppressed and induced in susceptible and resistant cultivars, and the finding that that *Z. tritici* suppresses the production of immune-related metabolites in a susceptible cultivar. This is an interesting finding, logic in my eyes and well known from other microbe-host interactions.

The strength of the study is the comprehensive experimental design including coinfection assays, comparative metabolomics, and microbiome profiling to study the interaction of *Z. tritici* in susceptible and resistant wheat as well as the logic, hypothesis-driven experimental design.

The weakness of the study is that this new plant-pathogen interaction was shown only this this artificial system for one pathogen, even for one strain of the pathogen. Here, it needs more evidence that it occurs in the field or works for other pathogens too.

Reply: We would like to point out to the reviewer that we do not try to claim that SIS is a phenomenon found in all fungal plant pathogen interactions. In fact, we underline that this phenomenon relates to this particular hemibiotrophic fungus during its infection of a susceptible host.

Our study has relied on the infection of the reference isolate IPO323. It would have been interesting to assess the impact of genetic variation among *Z. tritici* isolates on the wheat immune response. However, this has not been the aim of the present study. In our previous research we have investigated variation between isolates and we have noted great heterogeneity in the infection progress of isolates (Hauseisen *et al.*, Ecology and Evolution, 2019). The different infection programs of *Z. tritici* isolates make direct comparisons of transcriptome and metabolome data challenging. We have focused analyses in this study on IPO323 because:

1. Our previous research has demonstrated that different strains of *Z. tritici* (including the reference IPO323) differ dramatically in the timing of the infection stages although resulting in similar disease outcomes (Hauseisen *et al.*, 2019). For the co-infection experiments conducted here, it was crucial that the fungal infection was in the biotrophic phase. As the timing is so different between *Z. tritici* isolates we would have had to use different time points for every fungal isolate which could have introduced a considerable amount of bias in our assessment, such as leaf age.

2. IPO323 is the reference strain of *Z. tritici*. It is very common in phytopathology that experiments are carried out solely with the reference strain and conclusions concerning the pathosystem in general are drawn from a reference strain for which many resources and are available. In future studies we hope, however, to be able to address the role of genetic variation in *Z. tritici* during plant infection and SIS.

As mentioned, we appreciate the reviewer's critique and have changed text in the manuscript to point out that the strength of SIS may vary among isolates (see line 348/349).

The pathogen induced strong shifts in wheat leaf microbiota. It is of value that these studies are integrated, and the extend of the shift shows how important this is. However, as presented these results are isolated, and should be more linked with the other results.

Reply: We apologize if the connection between the microbiome study and the co-infection experiments have not been clearly formulated. For all three experimental approaches 1) phytopathology, 2) metabolomics and 3) microbiome, we aim to answer the same underlying question: What is the effect of *Z. tritici* on wheat immune responses. The experimental approaches each provide different "readouts". Importantly, for all experiments we follow the same experimental design as depicted in figure S3. This shared experimental design connects the three experimental approaches and the results obtained.

Furthermore, we consider the microbial profiling an "up-scaling" of the *Pseudomonas* experiments. Hereby we aim to describe the impact of fungal infection not only on one bacterial species, but on all members of the community. We thereby demonstrate how other microbiota members increase or decrease in relative abundance in Chinese Spring and Obelisk. Also, in the discussion we link certain metabolites identified by the metabolome analyses to their known effects on the microbiome. To address the point raised by the reviewer, we have modified the introduction of the experimental approaches to make the connections more obvious to the reader (for examples, see lines 125/126, 206 or 210/211).

Moreover, I would suggest, to look at the whole pathosystem from the microbiome perspective. The pathogen is a member of the microbiome as well.

Reply: For the microbiome study, we only conducted 16S sequencing and focused on bacterial members of the microbiome. Therefore *Z. tritici* as a fungal pathogen is not included in the microbiome study. It would be interesting to investigate how other fungal endophytes perform in the presence of *Z. tritici*. However, we decided against using ITS sequencing for fungal members of the community. This is because; our treatment involves the application of fungal spores of *Z. tritici*. This ITS amplicons would have been heavily enriched with *Z. tritici* sequences. Future studies with the aim of investigating fungal-fungal interactions in the phyllosphere should rely on another experimental design.

Qualitative data of the microbiota are missing!

Reply: We are unsure which type of qualitative data that the reviewer would like to see. In our Results section, we report on qualitative differences among microbiomes of the different wheat cultivars and treatments (see Figure 5). We have now included additional qualitative data on the microbiota (see Supplemental figure S12)

The definition of plant pathogens in these categories biotrophic, hemibiotrophic etc. is a really anthropocentric view, which needs revision. In this manuscript I would not place it so central that *Zymoseptoria* is hemibiotrophic. It's the function of all fungi to degrade organic matter.

Reply: The categories may reflect an anthropocentric but they are widely used in the fields of plant biology. In the literature *Z. tritici* is defined as a hemi-biotrophic pathogen. The conclusion we can draw from our experiments is that *Z. tritici* is not inactive before it switches to a necrotrophic stage. Often the early phase of infection is described as a latent phase. However, here we shown that the fungus is suppressing the plant immune system as it is known from “true” biotrophic pathogens. We therefore emphasize that the fungus is a hemibiotroph, although we acknowledge that the exact feeding strategy of the pathogen during this early phase of infection remains unclear. We note that the term ‘hemibiotrophic fungi’ also is described as fungi ‘that require living plant tissue to survive and complete their life cycle’ (Koeck *et al.*, Cellular Microbiology, 2011). We consider that this definition applies to *Z. tritici* and we would therefore like to keep the term in the manuscript.

The title has to be changed – it is too broad, only one pathosystem was studied.

Reply: The title “Hemibiotrophic fungal pathogen induces systemic susceptibility and systemic shifts in wheat metabolome and microbiome composition” has been changed to “Fungal pathogen induces systemic susceptibility and systemic shifts in wheat metabolome and microbiome composition”

Discussion section would profit from shortening.

Reply: We believe that our manuscript presents many results and that the Discussion section is important to interpret the different datasets in a common context. Therefore, we would like to keep the Discussion with the current content.

Reviewer #3 (Remarks to the Author):

The manuscript reports on the correlation between SAR and pathogenic fungi.

The study shows very interesting results, a possibility for fungi to suppress plant immune system based on metabolomics data.

I would like to recommend the publication of the manuscript.

One comment:

Line 123, 37,664 seems the number of MS peaks. Then, to say the number is for metabolites is rather doubtful. So, better to say it is the number of peaks or signals in the analytical system.

Reply: We would like to thank the reviewer for pointing out this inaccuracy in our terminology. In the revised manuscript, we now use the word “peaks” instead of metabolites to indicate that we found that large numbers of unknown metabolites or analytical signals (see lines 129, 150-159).

Reviewer #4 (Remarks to the Author):

The manuscript aims to investigate the interaction between the fungal pathogen *Z. tritici* and wheat. The work presents three related stories. In the first story, the authors showed that upon infection with the fungal pathogen *Zymoseptoria tritici* an induced susceptibility in adjacent regions of the leaf is triggered. Second, they showed that there are differences in the accumulation of metabolomes in two wheat cultivars, one susceptible and one resistant. The third story is based on a microbiome analysis of wheat leaves and demonstrates that in the resistant cultivar there is a reduction in the richness and changes in the structure of the bacterial community upon infection with the pathogen. The work presents a very original work and provides good quality data which is of significance to the field. However, the work lacks a general focus and sometimes it is difficult to understand the logic of the text. Although the three stories are novel and related, the connection between them is not clear in the manuscript.

Reply: We are sorry that the connection between the experimental approaches also have been unclear to reviewer 4. As outlined above in our reply to reviewer 2 we have modified the manuscript to make the connections more obvious to the reader (for examples, see lines 125/126, 206 or 210/211).

The authors have a very novel and interesting finding. They demonstrate that *Z. tritici* triggers induced susceptibility. They call this phenomenon "systemic induced susceptibility" (SIS). The main goal of the work is to characterize SIS, but the analysis performed does not provide evidences for the consequences of SIS in the microbiome or the metabolites involved in this phenomenon.

Reply: We added an analysis concerning the consequences of SIS on the level of metabolites to convince the reviewer of the observed SIS phenomenon (see Supplemental figure S10). The metabolome analysis showed differences in the amount of lignin precursors after fungal infection between the susceptible and the resistant cultivar. We have now conducted a new experiment to measure the lignin content of both cultivars before and after infection with *Z. tritici*. We find an increase in lignin content in the resistant cultivar in response to fungal infection. In contrast, lignin content decreases after fungal infection in the susceptible cultivar, providing additional evidence for the occurrence of SIS.

To achieve this, first of all, the comparisons in the metabolome should be performed between mock and infected plants. In the results and figures, it seems (although it is not very clear) that this is not the main comparison performed. Comparisons between the two cultivars is valid, but would not provide information on what are the compounds involved in SIS (and SAR). This is mainly due to the

complex genetic background of both wheat cultivars and that SIS (and SAR) are induced upon infection, as shown in figure 2.

Reply: We carried out three comparisons: 1) treatment, 2) cultivar, 3) position. The comparisons were carried out in an identical way, without preference to any of the comparisons. The “treatment” comparison revealed a lower number of significant metabolites compared to the other two comparisons. This is not surprising since the response to fungal infection should be very specific, whereas differences in cultivar and leaf position also can be influenced by the genetic background and developmental stage of the leaf. Nevertheless, all comparisons always included information about the respective treatment, whereby every significant metabolite can be associated either with fungal or mock infection.

In addition, changes in microbiome were only observed in the resistant cultivar, which points more towards the direction that SIS does not lead to changes in the microbiome. If there are evidences for SIS changing the microbiome, it should be explained in the text and figures. I suggest that either the focus of the manuscript switches towards the characterization of induced resistance by *Z. tritici* in wheat or that the data providing mechanistic understanding of SIS is highlighted.

Reply: We would like to thank the reviewer for this very interesting comment. Indeed, the changes in community richness in Obelisk between mock and Zt treatments are not antithetic to those in Chinese Spring. This does, however, not exclude the occurrence of SIS.

In Obelisk we observe that alpha diversity is not reduced upon fungal infection, as it is in Chinese Spring. However, our analyses of beta diversity still indicate that the microbial community also changes in Obelisk, although not as drastic as in Chinese Spring. It is likely that SIS as well as the presence of the fungus itself (including fungal produced metabolites) impact microbial community composition in Obelisk. We have corrected the manuscript to make this point clearer. In addition, we would like to draw the reviewer’s attention to figure S10 where we see a clear response of Obelisk to the fungal infection on the level of community structure.

More detailed comments can be found below:

1- The representation of the metabolome is not very clear. In figure 4b, it is very difficult to understand which comparison was performed based on the provided nomenclature. Based on the text, it seems that the main comparison was performed between both cultivars. If the hypothesis is that changes in the content of some metabolites lead to the observed phenotype upon infection by *Z. tritici*, then the authors should compare the metabolomic profile of infected and uninfected leaves. Most frequently the comparisons are between cultivars, which is also relevant, but not conclusive for SIS and SAR.

Reply: We understand the concern raised by the reviewer. We acknowledge the challenge of simplifying an overview of the metabolome analyses that include comparisons of different cultivars, treatments, time points and plant tissues. Figure S8c is intended to provide comprehensive overview of the nomenclature and terms used in our study.

As illustrated in Figure S8c, every comparison takes into account each of the given conditions i.e. information about the cultivar, the leaf area, and the treatment). For example, the comparison between Obelisk and Chinese Spring was conducted for adjacent leaf tissues in both cultivars that were treated with *Z. tritici*. In the comparison of cultivars we first clarify if the difference is present under mock conditions or with fungal infection.

Comparing Chinese Spring (CS) adjacent *Z. tritici* (Zt) and CS adjacent mock (as the reviewer suggested) will only reveal a difference in Chinese Spring. This may indeed be helpful to illustrate the occurrence of SAR. But with this comparison we do not demonstrate how this differs in the susceptible cultivar or between the two cultivars. The comparison between cultivars is however a main objective of these analyses. A detailed explanation of the content of the comparisons is included in the manuscript (see lines 142-146).

The authors had interesting results related to the content of BXs in resistant and susceptible cultivars. In the resistant cultivar they observed increased levels of the inactive forms in adjacent tissues of the infection by the fungus. However, this was not observed in the susceptible cultivar. The authors conclude from this data that the fungus might manipulate the host in order to prevent the induction of this compound. However, it could very well be that only in Chinese Spring these inactive forms are accumulated and not in Obelisk. Actually, this seems to be the case since in Fig. S9 there is no induction of the BXs, but CS has higher constitutive levels of them. Thus, their function in SIS remains unclear and *Z. tritici* does not seem to have capacity to suppress their accumulation.

Reply: Indeed, Chinese Spring constitutively produces high levels of several components of the BXs biosynthesis pathway. We have included figure 9S to the manuscript to make this visible. High constitutive levels do not exclude that the metabolite production can be further increased upon *Z. tritici* infection. We see an induction of BX upon *Z. tritici* infection, e.g. for HMBOA-Gly locally at 4dpi and in adjacent tissue at 4 and 8dpi (Figure S9).

In contrast, we see a reduction in BX levels upon fungal infection in Obelisk, e.g. HBOA, HBOA-Gly, HMBOA and HMBOA-Gly locally at 4dpi, pointing towards an inhibition of these compounds by *Z. tritici*. We did not find examples in Obelisk where BX compounds synthesized downstream of HBOA were induced by the fungus. Although we could not detect some of the compounds in Obelisk, we cannot exclude their presence at concentrations that are below our detection limit. We do however see no reason to suspect this and can at the same time not rule out that there is a likewise suppression of these compounds on a lower level.

In addition, it is known that Chinese Spring basal resistance is not higher than in other cultivars, but this cultivar is specifically able to recognize an effector of some strains of *Z. tritici* and subsequently trigger an immune response that hinders the progression of the pathogen (by the way this is key information that should be described in the introduction).

Reply: In the revised manuscript we have included this relevant information in the text (previously reference 18, now reference 20, line 76). Furthermore, the paragraph is substantially re-written to include the issues raised here by the reviewer (see lines 78-81).

The accumulation of the metabolites investigated in this work could contribute to the resistance response, but if this is the case, there should be differences in CS between the Mock and Zt treatments in figure 3c,d and S9. Based on the provided data this does not seem the case.

Reply: As explained above, there are differences between mock and *Z. tritici*-treated samples. In Figure 3 these differences are not demonstrated, but they are illustrated in Figure S9.

In summary, I think that the authors should focus on differentially accumulated compounds upon infection in order to have more conclusive data on SIS (and also on SAR).

Reply: For every DAM that is mentioned in the manuscript we have also provided information about the treatment (mock or *Z. tritici*-infected). In our response above, we have listed our arguments why we think the treatment and the cultivar comparisons are equally important.

2- In the discussion, the authors indicate that the data showed that pathogens have an active role in manipulating the microbiome (lines 316-322) in a susceptible host. Although this is true for the co-infections performed with *Pseudomonas*, it is not the case for the global microbiome analysis performed, since there are no differences between mock and Zt treatments in cultivar Obelisk in the figure 5.

The main goal is to characterize the effect of SIS in the microbiome and based on the data presented, there is an effect on the bacterial community upon a resistance response against *Z. tritici*, but not in the compatible interaction. If there were observed changes in the susceptible cultivar, they should be shown in the results.

Reply: As mentioned above, we believe that the absence of different types of resistance responses in Obelisk reflects an inhibition of immune responses in this susceptible cultivar. We show that up-regulation of immune-related metabolites coincide with a reduction in microbial diversity suggesting that a fraction of the phyllosphere microbiota is unable to tolerate the immune-related metabolites. In Obelisk we show that microbial richness is little affected. However, the community composition changes slightly. We consider that the presence of the fungus has an effect on the microbiome composition mostly likely conferred by SIS or by fungal produced metabolites. We have clarified these points in the manuscript (see lines 206 and 229/230). The microbiota change is furthermore illustrated in figure S10.

3- Finally, although in the results it is said that there are statistical differences in figure 6b and 6d at 16 dpi, it is not clear in the figure. Either the asterisks are missing or they are difficult to see. In case that there are no statistical differences, it cannot be concluded that induced susceptibility in systemic leaf tissues is a mechanism to promote further infections by *Z. tritici* and also the results section should be modified.

Reply: We do not report statistically significant differences in Figure 6 in the manuscript, because the observed differences are non-significant. We do however see a tendency that supports our hypothesis. The challenge when interpreting these results is that the last timepoint for readout was

conducted at 16 dpi. Hereby, we have not extended the experiment to the final stage of fungal sporulation, ca 21 dpi. The trend that we show in Figure 6 however indicates that differences in asexual sporulation and necrosis would be even larger at later stages of infection.

We note here that this final experiment was conducted to test a hypothesis that emerged from the main result of the study (the occurrence of SIS). We aimed to address why the fungus so strongly suppresses the immune system of the host. We think it is to promote the own propagation from one leaf to the next. But this question was not the main one raised in our study.

4- I think that there are some sentences that are not accurate in the discussion/results. For instance, the authors mentioned that the role of fungal effector proteins in SIS is demonstrated in the experiment of T3SS. For making this statement, first the authors need to clarify if the *hrcC* mutant grows more in the presence of the fungus. Based on the figure S5, there are no statistical differences between mock "Pst hcC" and "Zt Pst hrcC". Therefore, the sentences "the increased growth of Pst hrcC in *Z. tritici*-infected Obelisks..." (line 254) and "Coinfection with *Z. tritici* also enabled growth of the T3SS-defective..." (line 110) are not correct. Even if this is the case, this experiment only highlights that effectors of *Pst* are not required, most probably because of a suppression of the immune response by *Z. tritici*, but it is not suggesting anything about the role of *Z. tritici* effectors in suppressing the immune response of wheat.

Reply: As indicated in figure S5, the statistical difference between "Pst hcC" and "Zt Pst hrcC" is not significant (p -value = 0.07258). We have rephrased the respective sentences with a more careful interpretation of the results (see lines 115 and 276). We would however like to point out that it is very unlikely that *Z. tritici* is suppressing the wheat immune system with something else than effector proteins or non-proteinaceous effectors like small RNAs or metabolites.

Minor comments:

- The results shown in figure 1a and 1b are already known in the literature, but it was not cited properly. The authors should highlight that this was already known and cite Brown et al. 2001.

Reply: As mentioned above at an earlier reply to this reviewer, we have cited Brown et al 2001 (previously reference 18, now reference 20) when discussing the result of figure 1a and 1b. We have improved this section to make sure the citation does not go unnoticed (see lines 78-83).

- The title of figure S5 is very difficult to understand.

Reply: We have improved the title of figures S5

- In figure S7 there is a typo. It should say Obelisk.

Reply: We have corrected the typo in Figure S7 to Obelisk

- In the metabolome section, it is not clear if the analysis was performed for the annotated or for the global metabolome. In line 125 it is mentioned that only annotated metabolites will be analyzed, but in line 126, it is mentioned that the complete metabolome dataset will be analysed.

Reply: We first conducted a general analysis of the complete dataset (line 126 to 132). Thereafter, we focused on annotated metabolites. We have clarified this in the text (see lines 132/133 and 141).

- In this analysis, it is striking to observe that there are differences in the mock between local and adjacent. Do the authors have any explanation for that?

Reply: We are aware of these differences between local and adjacent leaf tissue in the metabolomic study. We included the 'position' comparison to indicate these differences. We believe that we included all the necessary controls to rule out that the given differences between local and adjacent tissue could bias our conclusions. Differences in metabolite composition in local and adjacent leaf tissue can probably be explained by the following factors:

1. In contrast to the adjacent tissue, local tissue has been mock treated. Since mock treatment is also a treatment that physically disturbs the leaf surface it may confer differences in treated versus untreated tissue.
2. Wheat leaves expand from the leaf base towards the tip. This means that tissue closer to the leaf tip is older than tissue to the leaf base. Differences in metabolite composition could therefore also be caused by differences in tissue age.

- Figure 3c,d would be clearer if it will also indicate which are the active and inactive forms.

Reply: We have modified Figure 3 to differentiate between active and inactive form of BXs.

- In line 162, it is mentioned "We suspect that the absence of BXs in Obelisk is partly responsible for the increased susceptibility of this cultivar". However, since we know that the resistance of CS is due to the immune response triggered upon recognition of an effector protein, it is most probably the other way round that the presence of BXs is responsible for the resistance of Chinese Spring.

Reply: We agree with the reviewer that BXs play a role in the resistance of Chinese Spring towards *Z. tritici* but is not solely responsible. Nevertheless, we still consider that the absence of BXs in Obelisk may contribute to its susceptibility towards *Z. tritici*. We have clarified this point in the manuscript (for example, see lines 171-173 and 229/230).

Related to this, it is mentioned that BXs are not accumulated in cv. Obelisk (line 161), but this is not what is shown in figure S9.

Reply: Figure S9 shows that there is no accumulation of BXs upon fungal infection in Obelisk. We have clarified this point in the manuscript (line 171).

In any case, in order to conclude about SIS and SAR, the authors should compare infected and non-infected tissues. Only upon infection, they observed differential colonization by the bacteria. Thus, if these metabolites are having a role, they should be differentially accumulated upon infection and not between cultivars.

Reply: Please see our explanation above on the three comparisons, which we carried out (see page 6 and page 8/9 of this document)

- The sentence written in lines 183 and 184 is not very accurate, since there are not enough evidences to highlight that *Z. tritici* manipulates immune-related biosynthetic pathways in susceptible wheat cultivars.

Reply: We have clarified the respective sentence (now lines 202/203).

- The statistical analysis was performed correctly, but I miss information on how many times were the experiments performed independently. Specially for the infection assays.

Reply: The information has been added to the manuscript (see figure legends).

- Line 409 in materials and methods. How were the seeds washed?

Reply: Seeds were washed 3 times (1 min each) with 20 mL sterile water, then 1% phosphate-buffered saline (v/v, PBS) supplemented with 0.02% (v/v) Tween 20, and then sterile 1% PBS. Two seeds per cultivar (i.e., *T. aestivum* cultivar Chinese Spring or Obelisk) were sowed in a pot containing 155 g of peat (HAWITA GmbH, Vechta, Germany). This information is indicated in Supplementary materials line 153-156.

- The results section has frequently information that should have been provided in the introduction section.

Reply: In order to keep the introduction short and general, specific information is given in the result section wherever needed. If the editor objects to this style, we will rewrite the introduction upon request.

REVIEWERS' COMMENTS:

Reviewer #4 (Remarks to the Author):

The work presents very novel and very important findings. However, there are some analysis that are not clear and conclusions that are still not supported by the results shown.

The main reason for this comment is that with the analysis presented we cannot differentiate between specific induced immunity and induced susceptibility. Chinese Spring is able to specifically recognize one avirulence factor of IPO323 and induce an immune response suggesting that the observed differences in accumulation of metabolites between the cultivars is due to the immune response induced in Chinese Spring and not to the induced susceptibility induced in Obelisk (As now indicated in lines 173-174). Having a strong conclusion on the fact that *Z. tritici* manipulates the accumulation of compounds involved in plant defense responses (as indicated in lines 198-200) is more challenging and I do not think that the provided data demonstrate this. However, I agree that the results presented do not contradict this statement and actually go in the good direction. The new experiment included in figure S10 is a good example in which *Z. tritici* might suppress the accumulation of a compound (lignin) since there are differences in the accumulation pattern in Obelisk between the infected and not infected plants compared to Chinese Spring. I think that this kind of comparisons, which incorporate both the treatment and the cultivar, are the ones that need to be performed in order to demonstrate the role of metabolites in SIS. And not to compare for each treatment or position the accumulation of a certain compound between both cultivars. If there are other compounds with a similar accumulation pattern as the one shown in Figure S10, it will provide more evidences for SIS and the role of metabolites.

About figure 6, still is not clear how many times was the experiment performed. If it was only performed once, a biological replicate will be needed. I agree that there might have been differences at 21 dpi, but extrapolations should not be made with infection assays and the results presented do not show differences between both treatments. Thus, the sentence in lines 254-255 ("fungal success on the systemic leaf was increased with infection on the second leaf at 16 dpi") is not correct.

REVIEWERS' COMMENTS:

Reviewer #4 (Remarks to the Author):

The work presents very novel and very important findings. However, there are some analysis that are not clear and conclusions that are still not supported by the results shown. The main reason for this comment is that with the analysis presented we cannot differentiate between specific induced immunity and induced susceptibility. Chinese Spring is able to specifically recognize one avirulence factor of IPO323 and induce an immune response suggesting that the observed differences in accumulation of metabolites between the cultivars is due to the immune response induced in Chinese Spring and not to the induced susceptibility induced in Obelisk (As now indicated in lines 173-174).

Reply: We very much appreciate the reviewer's efforts of checking the conclusions in detail. In line 192-193 (formerly 172-174) we say that we "suspect" the difference in immune related metabolites "contributes" to the different resistance phenotypes. We think this is a careful statement and hope the reviewer can agree with this given the additional changes in the text (see reply to the editorial requests and below).

Having a strong conclusion on the fact that *Z. tritici* manipulates the accumulation of compounds involved in plant defense responses (as indicated in lines 198-200) is more challenging and I do not think that the provided data demonstrate this. However, I agree that the results presented do not contradict this statement and actually go in the good direction. The new experiment included in figure S10 is a good example in which *Z. tritici* might suppress the accumulation of a compound (lignin) since there are differences in the accumulation pattern in Obelisk between the infected and not infected plants compared to Chinese Spring. I think that this kind of comparisons, which incorporate both the treatment and the cultivar, are the ones that need to be performed in order to demonstrate the role of metabolites in SIS. And not to compare for each treatment or position the accumulation of a certain compound between both cultivars. If there are other compounds with a similar accumulation pattern as the one shown in Figure S10, it will provide more evidences for SIS and the role of metabolites.

Reply: In the newly revised manuscript we have not included new additional experimental data. We would like to point out that the lignin data did not derive from the metabolomics data but was conducted independently, building on the observation that lignin precursors accumulated in infected Chin. Spring samples in the metabolomics study. We agree with the reviewer that future experiments are important to validate the observed differences from the metabolome data with biochemical assays. However, we are currently not aware of an additional extraction method comparable to the one with lignin that would allow us to specifically isolate certain metabolites (or metabolite groups as e.g. the HCAAs) from leaf samples to compare the accumulation in the two wheat cultivars upon fungal infection. Approaches of testing the functional role of single metabolites identified in the metabolomics study to be involved in SIS are outside of the scope of this manuscript. We have however made several changes in the manuscript to ensure that we only present the observed patterns as correlation and not causalities, for example in lines 45-46, 220, 235, 267 and others. All changes are highlighted in the word document with shown track-changes.

About figure 6, still is not clear how many times was the experiment performed. If it was only performed once, a biological replicate will be needed. I agree that there might have been differences at 21 dpi, but extrapolations should not be made with infection assays and the

results presented do not show differences between both treatments. Thus, the sentence in lines 254-255 ("fungal success on the systemic leaf was increased with infection on the second leaf at 16 dpi") is not correct.

Reply: Already in the title we only say that "SIS may promote" fungal infection on the systemic leaf. We are aware and now also clearly state in the text (line 275) as well as the figure that the present results do not show significant differences. We do however think that the experiment gives the reader an important perspective and contributes to the biological relevance of SIS from the perspective of Z. tritici. We therefore would like to keep the figure in the current state and hope the reviewer can agree with the changes we applied.